



# The Monash Simple Climate Model Experiments (MSCM-DB v1.0): An interactive database of mean climate, climate change and scenario simulations

By Dietmar Dommenget[1*], Kerry Nice[1,4], Tobias Bayr[2], Dieter Kasang[3], Christian Stassen[1] and Mike Rezny[1]

*: corresponding author; dietmar.dommenget@monash.edu
1: Monash University, School of Earth, Atmosphere and Environment, Clayton, Victoria 3800, Australia.
2: GEMOAR Helmholtz Centre for Ocean Research, Düsternbrooker Weg 20, 24105 Kiel, Germany
3: DKRZ, Hamburg, Germany
4: Transport, Health, and Urban Design Hub, Faculty of Architecture, Building, and Planning, University of Melbourne, Victoria 3010, Australia

submitted the Geoscientific Model Development, 8 March 2018

## Abstract

This study introduces the Monash Simple Climate Model (MSCM) experiment database. The model simulations are based on the Globally Resolved Energy Balance (GREB) model. They provide a basis to study three different aspects of climate model simulations: (1) understanding the processes that control the mean climate, (2) the response of the climate to a doubling of the $CO_2$ concentration, and (3) scenarios of external $CO_2$ concentration and solar radiation forcings. A series of sensitivity experiments in which elements of the climate system are turned off in various combinations are used to address (1) and (2). This database currently provides more than 1,300 experiments and has an online web interface for fast analysis of the experiments and for open access to the data. We briefly outline the design of all experiments, give a discussion of some results, and put the findings into the context of previously published results from similar experiments. We briefly discuss the quality and limitations of the MSCM experiments and also give an outlook on possible further developments. The GREB model simulation of the mean climate processes is quite realistic, but does have uncertainties in the order of 20-30%. The GREB model without flux corrections has a root mean square error in mean state of about 10°C, which is larger than those of general circulation models (2°C). However, the MSCM experiments show good agreement to previously published studies. Although GREB is a very simple model, it delivers good first-order



estimates, is very fast, highly accessible, and can be used to quickly try many
different sensitivity experiments or scenarios.

## 1. Introduction

Our understanding of the dynamics of the climate system and climate changes is
strongly linked to the analysis of model simulations of the climate system using a
range of climate models that vary in complexity and sophistication. Climate
model simulations help us to predict future climate changes and they help us
gain a better understand of the dynamics of this complex system.
State-of-the-art climate models, such as used in the Coupled Model Inter-
comparison Project (CMIP; Taylor et al. 2012), are highly complex simulations
that require significant amounts of computing resources and time. Such model
simulations require a significant amount of preparation. The development of
idealized experiments that would help in the understanding and modelling of
climate system processes are often difficult to realize with the complex CMIP-
type climate models. In this context, simplified climate models are useful, as they
provide a fast first guess that help to inform more complex models. They also
help in understanding the interactions in the complex system.
In this article, we introduce the Monash Simple Climate Model (MSCM) database
(version: MSCM-DB v1.0). The MSCM is an interactive website
(http://mscm.dkrz.de, Germany and http://monash.edu/research/simple-
climate-model, Australia) and database that provide access to a series of more
than 1,300 experiments with the Globally Resolved Energy Balance (GREB)
model [Dommenget and Floter 2011; here after referred to as DF11]. The GREB
model was primarily developed to conceptually understand the physical
processes that control the global warming pattern in response to an increase in
$CO_2$ concentration. It therefore centres around the surface temperature ($T_{surf}$)
tendency equation and simulates only the processes needed for resolving the
global warming pattern.
Simplified climate models, such as Earth System Models of Intermediate
Complexity (EMICs), often aim at reducing the complexity to increase the
computation speed and therefore allow faster model simulations (e.g. CLIMBER
[Petoukhov et al. 2000], UVic [Weaver et al. 2001], FAMOUS [A] or LOVECLIM
[Goosse et al. 2010]). These EMICs are very similar in structure to state-of-the-
art Coupled General Circulation Models (CGCMs), following the approach of
simulating the geophysical fluid dynamics. The GREB model differs, in that it
follows an energy balance approach and does not simulate the geophysical fluid
dynamics of the atmosphere. It is therefore a climate model that does not include
weather dynamics, but focusses on the long term mean climate and its response
to external boundary changes.
The purpose of the MSCM database for research studies are the following:

• *First Guess*: The MSCM provides first guesses for how the climate may
change in idealized or realistic experiments. The MSCM experiments can
be used to test ideas before implementing and testing them in more
detailed CGCM simulations.

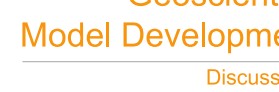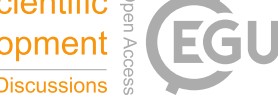

• **Null Hypothesis**: The simplicity of the GREB model provides a good null
hypothesis for understanding the climate system. Because it does not
simulate weather dynamics or circulation changes of neither large nor
small scale it provides the null hypothesis of a climate as a pure energy
balance problem.
• **Conceptual understanding:** The simplicity of the GREB model helps to
better understand the interactions in the complex climate and, therefore,
helps to formulate simple conceptual models for climate interactions.
• **Education**: Studying the results of the MSCM helps to understand the
interactions that control the mean state climate and its regional and
seasonal differences. It helps to understand how the climate will respond
to external forcings in a first-order approximation.
The MSCM provides interfaces for fast analysis of the experiments and selection
of the data (see Figs. 1-3). It is designed for teaching and outreach purposes, but
also provides a useful tool for researchers. The focus in this study will be on
describing the research aspects of the MSCM, whereas the teaching aspects of it
will not be discussed.  The MSCM experiments focus on three different aspects of
climate model simulations: (1) understanding the processes that control the
mean climate, (2) the response of the climate to a doubling of the $CO_2$
concentration, and (3) scenarios of external $CO_2$ concentration and solar
radiation forcings. We will provide a short outline of the design of all
experiments, give a brief discussion of some results, and put the findings into
context of previously published literature results from similar experiments.
The DF11 study focussed primarily on the development of the model equations
and the discussion of the response pattern to an increase in $CO_2$ concentration.
This study here will give a more detailed discussion on the performance of the
GREB model on simulation of the mean state climate.
The paper is organized as follows: The following section describes the GREB
model, the experiment designs, the MSCM interface, and the input data used. A
short analysis of the experiments is given in section 3. This section will mostly
focus on the GREB model performance in comparison to observations and
previously published simulations in the literature, but it will also give some
indications of the findings in the model experiments and the limitations of the
GREB model. The final section will give a short summary and outlook for
potential future developments and analysis.
## 2. Model and experiment descriptions
The GREB model is the underlying modelling tool for the MSCM interface. The
development of the model and all equations have been presented in DF11. The
model is simulating the global climate on a horizontal grid of 3.75° longitude x
3.75° latitude and in three vertical layers: surface, atmosphere and subsurface
ocean. It simulates the main physical processes that control the surface
temperature tendencies: solar (short-wave) and thermal (long-wave) radiation,
the hydrological cycle (including evaporation, moisture transport and
precipitation), horizontal transport of heat and heat uptake in the subsurface
ocean. Atmospheric circulation and cloud cover are seasonally prescribed
boundary condition, and state-independent flux corrections are used to keep the

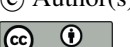



GREB model close to the observed mean climate. Thus, the GREB model does not
simulate the atmospheric or ocean circulation and is therefore conceptually very
different from CGCM simulations.
The model does simulate important climate feedbacks such as the water vapour
and ice-albedo feedback, but an important limitation of the GREB model is that
the response to external forcings or model parameter perturbations do not
involve circulation or cloud feedbacks, which are relevant in CGCM simulations
[Bony et al. 2006].
Input climatologies (e.g. $T_{surf}$ or atmospheric humidty) for the GREB model are
taken from the NCEP reanalysis data from 1950-2008 [Kalnay et al. 1996], cloud
cover climatology from the ISCCP project [Rossow and Schiffer 1991], ocean
mixed layer depth climatology from Lorbacher et al. [2006], and topographic
data was taken from ECHAM5 atmosphere model  [Roeckner et al. 2003].
GREB does not have any internal (natural) variability since daily weather
systems are not simulated. Subsequently, the control climate or response to
external forcings can be estimated from one single year. The primary advantage
of the GREB model in the context of this study is its simplicity, speed, and low
computational cost. A one year GREB model simulation can be done on a
standard PC computer in about 1 s (about 100,000 simulated years per day). It
can do simulations of the global climate much faster than any state-of-the-art
climate model and is therefore a good first guess approach to test ideas before
they are applied to more complex CGCMs. A further advantage is the lag of
internal variability which allows the detection of a response to external forcing
much more easily.
**a.  Experiments for the mean climate deconstruction**
The conceptual deconstruction of the GREB model to understand the interactions
in the climate system that lead to the mean climate characteristics is done by
defining 11 processes (switches; see Fig. 1). For each of these switches, a term in
the model equations is set to zero or altered if the switch is "OFF". The processes
and how they affect the model equations are briefly listed below (with a short
summary in Table 1). The model equations relevant for the experiments in this
study are briefly restated in the appendix section A1 for the purpose of
explaining each experimental setup in the MSCM.
**Ice-albedo**: The surface albedo ($\alpha_{surf}$) and the heat capacity over ocean points
($\gamma_{surf}$) are influenced by snow and sea ice cover. In the GREB model these are a
direct function of $T_{surf}$. When the Ice-albedo switch is OFF the surface albedo of
all points is constant (0.1) and, for ocean points, $\gamma_{surf}$ follows the prescribed
ocean mixed layer depth independent of $T_{surf}$ (i.e. no ice-covered ocean).
**Clouds**: The cloud cover, *CLD*, influences the amount of solar radiation absorbed
at the surface ($\alpha_{clouds}$ in eq. [A5]) and the emissivity of the atmospheric
layer, $\varepsilon_{atmos}$, for thermal radiation (eq. [A8]). When the Clouds switch is OFF, the
cloud cover is set to zero.
**Oceans:** The ocean in the GREB simulates subsurface heat storage with the
surface mixed layer (~upper 50-100m). When the ocean switch is OFF, the $F_{ocean}$

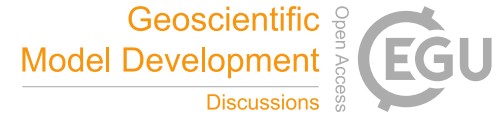



term in eq. [A1] is set to zero, eq. [A3] is set to zero and the heat capacity off all
ocean points is set to that of land points.

**Atmosphere**: The atmosphere in the GREB model simulates a number of
processes: The hydrological cycle, horizontal transport of heat, thermal
radiation, and sensible heat exchange with the surface. When the atmosphere
switch is OFF, eq. [A2] and [A4] are set to zero, the heat flux terms, $F_{sense}$ and
$F_{latent}$ in eq. [A1] are set to zero and the downward atmospheric thermal
radiation term in eq. [A6] is set to zero.

**Diffusion of Heat**: The atmosphere transports heat by isotropic diffusion (4[th]
term in eq. [A2]). When this process is switched OFF, the term is set to zero.

**Advection of Heat**: The atmosphere transports heat by advection following the
mean wind field, $\vec{u}$ (5[th] term in eq. [A2]). When this process is switched OFF, the
term is set to zero.

**CO₂:** The $CO_2$ concentration affects the emissivity of the atmosphere, $\varepsilon_{atmos}$ (eq.
[A9]). When this process is switched OFF, the $CO_2$ concentration is set to zero.

**Hydrological cycle**: The hydrological cycle in the GREB model simulates the
evaporation, precipitation, and transport of atmospheric water vapour. It further
simulates latent heat cooling at the surface and heating in the atmosphere. When
the hydrological cycle is switched OFF, eq. [A4] is set to zero, the heat flux term
$F_{latent}$ in eq. [A1] is set to zero, and $viwv_{atmos}$ in eq. [A9] is set to zero.
Subsequently, atmospheric humidity is zero.
It needs to be noted here, that the atmospheric emissivity in the log-function
parameterization of eq. [A9] can become negative, if the hydrological cycle, cloud
cover and $CO_2$ concentration are switched OFF (set to zero). This marks an
unphysical range of the GREB emissivity function and we will discuss the
limitations of the GREB model in these experiments in Section 3b.

**Diffusion of Water Vapour**: The atmosphere transports water vapour by
isotropic diffusion (3[rd] term in eq. [A4]). When this process is switched OFF, the
term is set to zero.

**Advection of Water Vapour**: The atmosphere transports heat by advection
following the mean wind field, $\vec{u}$ (5[th] term in eq. [A2]). When this process is
switched OFF, the term is set to zero.

**Model Corrections**: The model correction terms in eqs. [A1, A3 and A4]
artificially force the mean $T_{surf}$, $T_{atmos}$, and $q_{air}$ climate to be as observed. When
the model correction is switched OFF, the three terms are set to zero. This will
allow the GREB model to be studied without any artificial corrections and
therefore help to evaluate the GREB model equations' skill in simulating the
climate dynamics.
It should be noted here that the model correction terms in the GREB model have
been introduced to study the response to doubling of the $CO_2$ concentration for
the current climate, which is a relative small perturbation if compared against

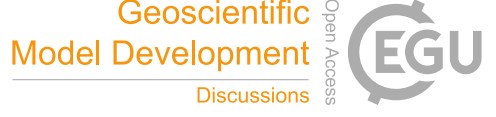



the other perturbations considered above. They are meaningful for a small
perturbation in the climate system, but are less likely to be meaningful when
large perturbations to the climate system are done (e.g. cloud cover set to zero).
Each different combination of the above-mentioned process switches defines a
different experiment. However, not all combinations of switches are possible,
because some of the process switches are depending on each other (see Table 1
and Fig. 1). The total number of experiments possible with these process
switches is 656. For each experiment, the GREB model is run for 50 years,
starting from the original GREB model climatology and the final year is
presented as the climatology of this experiment in the MSCM database.
**b.  Experiments for the 2x$CO_2$ response deconstruction**
The conceptual deconstruction of the GREB model to understand the interactions
in the climate system that lead to the climate response to a doubling of the $CO_2$
concentration can be done in a similar way, as described above for the mean
climate.  However, there are a number of differences that need to be considered.
A meaningful deconstruction of the response to a doubling of the $CO_2$
concentration should consider the reference control mean climate since the
forcings and the feedbacks controlling the response are mean state dependent.
We therefore ensure that all sensitivity experiments in this discussion have the
same reference mean control climate. This is achieved by estimating the flux
correction term in eqs. [A1, A3 and A4] for each sensitivity experiment to
maintain the observed control climate. Thus, when a process is switched OFF, the
control climatological tendencies in eqs. [A1, S3 and S4] are the same as in the
original GREB model, but changes in the tendencies due to external forcings, such
as doubling of the $CO_2$ concentration are not affected by the disabled process.
This is the same approach as in DF11.
For the 2x$CO_2$ response deconstruction experiments, we define 10 boundary
conditions or processes (switches; see Fig. 2). The Ice-albedo, advection and
diffusion of heat and water vapour, and the hydrological cycle processes are
defined in the same way as for the mean climate deconstruction (section 2a). The
remaining boundary conditions and processes are briefly listed below (and a
short summary is given in Table 2).
The following boundary conditions are considered:
**Topography**: The topography in the GREB model affects the amount of
atmosphere above the surface and therefore affects the emissivity of the
atmosphere in the thermal radiation (eq. [A9]). Regions with high topography
have less $CO_2$ concentration in the thermal radiation (eq. [A9]). When the
topography is turned OFF, all points of the GREB model are set to sea level height
and have the same amount of $CO_2$ concentration in the thermal radiation (eq.
[A9]).
**Clouds**: The cloud cover in the GREB model affects the incoming solar radiation
and the emissivity of the atmosphere in the thermal radiation (eq. [A9]). In
particular, it influences the sensitivity of the emissivity to changes in the $CO_2$
concentration. A clear sky atmosphere is more sensitive to changes in the $CO_2$



concentration than a fully cloud-covered atmosphere. When the cloud cover
switch is OFF, the observed cloud cover climatology boundary conditions are
replaced with a constant global mean cloud cover of 0.7. It is not set to zero to
avoid an impact on the global climate sensitivity, and to focus on the regional
effects of inhomogeneous cloud cover.
**Humidity**: Similarly, to the cloud cover, the amount of atmospheric water
vapour affects the emissivity of the atmosphere in the thermal radiation and, in
particular, the sensitivity to changes in the $CO_2$ concentration (eq. [A9]). A humid
atmosphere is less sensitive to changes in the $CO_2$ concentration than a dry
atmosphere. When the humidity switch is OFF, the constraint to the observed
humidity climatology (flux correction in eq. [A4]) is replaced with a constant
global mean humidity of 0.0052 [kg/kg]. It is again not set to zero to avoid an
impact on the global climate sensitivity, but to focus on the regional effects of
inhomogeneous humidity.
The additional feedbacks and processes considered are:
**Ocean heat uptake**: The ocean heat uptake in GREB is done in two ocean layers.
The largest part of the ocean heat is in the subsurface layer, $T_{ocean}$ (eq. [A3]).
When the ocean switch is OFF the $F_{ocean}$ term in eq. [A1] is set to zero, equation
[A3] is set to zero and the heat capacity ($\gamma_{surf}$) off all ocean points in eq. [A1] is
set to that of a 50m water column.
The total number of experiments with these process switches is 640. For each
experiment, the GREB model is run for 50 years, starting from the original GREB
model climatology and the changes relative to the original GREB model
climatology of this experiment is presented in the MSCM database.
**c. Scenario experiments**
A number of different scenarios of external boundary condition changes exist in
the MSCM experiment database. They include different changes in the $CO_2$
concentration and in the incoming solar radiation. A complete overview is given
in Table 3. A short description follows below.
**RCP-scenarios**
In the Representative Concentration Pathways (RCP) scenarios the GREB model
is forced with time varying $CO_2$ concentrations. All five different simulations have
the same historical time evolution of $CO_2$ concentrations starting from 1850 to
2000, and from 2001 follow the RCP8.5, RCP6, RCP4.5, RCP2.6 and the A1B $CO_2$
concentration pathways until 2100 [van Vuuren et al. 2011].
**Idealized $CO_2$ scenarios**
The 15 idealized $CO_2$ concentration scenarios in the MSCM experiment database
focus on the non-linear time delay and regional differences in the climate
response to different $CO_2$ concentrations. These were implemented in five
simulations in which the control $CO_2$ concentration (340ppm) was changed in
the first time step to a scaled $CO_2$ concentration of 0, 0.5, 2, 4, and 10 times the





control level. The 0.5x$CO_2$ and 2x$CO_2$ simulations are 50yrs long and the others
are 100yrs long.
Two different simulations with idealized time evolutions of $CO_2$ concentrations
are conducted to study the time delay of the climate response. In one simulation,
the $CO_2$ concentration is doubled in the first time step, held at this level for 30yrs
then returned to control levels instantaneously. In the second simulation, the $CO_2$
concentration is varied between the control and 2x$CO_2$ concentrations following
a sine function with a period of 30yrs, starting at the minimum of the sine
function at the control $CO_2$ concentration. Both simulations are 100yrs long.
The third set of idealized $CO_2$ concentration scenarios double the $CO_2$
concentrations restricted to different regions or seasons. The eight regions and
seasons include: the Northern or Southern Hemisphere, tropics (30ºS-30ºN) or
extra-tropics (poleward of 30º), land or oceans and in the month October to
March or in the month April to September. Each experiment is 50yrs long.
**Solar radiation**
Two different experiments with changes in the solar constant were created. In
the first experiment, the solar constant is increased by about 2% (+27W/m$^2$),
which leads to about the same global warming as a doubling of the $CO_2$
concentration [Hansen et al. 1997]. In the second experiment, the solar constant
oscillates at an amplitude of 1W/m$^2$ and a period of 11yrs, representing an
idealized variation of the incoming solar short wave radiation due to the natural
11yr solar cycle [Willson and Hudson 1991]. Both experiments are 50yrs long.
**Idealized orbital parameters**
A series of five simulations are done in the context of orbital forcings and the
related ice age cycles. In one simulation, the incoming solar radiation as function
of latitude and day of the year was changed to its values as it was 231Kyrs ago
[Berger and Loutre 1991 and Huybers 2006]. In an additional simulation, the $CO_2$
concentration is reduced from 340ppm to 200ppm as observed during the peak
of ice age phases in combination with the incoming solar radiation changes. Both
simulations are 100yrs long.
In three sensitivity experiments, we changed the incoming solar radiation
according to some idealized orbital parameter changes to study the effect of the
most important orbital parameters. The orbital parameters changed are: the
distance to the sun, the Earth axis tilt relative to the Earth-Sun plane (obliquity)
and the eccentricity of the Earth orbit around the sun. The orbit radius was
changed from 0.8AU to 1.2AU in steps of 0.01AU, the obliquity from -25° to 90° in
steps of 2.5° and the eccentricity from 0.3 (Earth closest to the sun in July) to 0.3
(Earth furthest from the sun in July) in steps of 0.01. Each sensitivity experiment
was started from the control GREB model (1AU radius, 23.5º obliquity and 0.017
eccentricity) and run for 50yrs. The last year of each simulation is presented as
the estimate for the equilibrium climate.
## 3. Some results of the model simulations
The MSCM experiment database includes a large set of experiments that address
many different aspects of the climate. At the same time, the GREB model has
limited complexity and not all aspects of the climate system are simulated in the

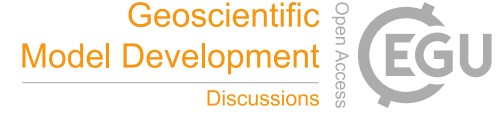



GREB experiments. The following analysis will give a short overview of some of the results that can be taken from the MSCM experiments. In this we will focus on aspects of general interest and on comparing the outcome to results of other published studies to illustrate the strength and limitations of the GREB model in this context. The discussion, however, will be incomplete, as there are simply too many aspects that could be discussed in this set of experiments. We will therefore focus on a general introduction and leave space for future studies to address other aspects.

**a. GREB model performance**

The skill of the GREB model is illustrated in Figure 4, by running the GREB model without the correction terms. For reference, we compare this GREB run with the observed mean climate and seasonal cycle (this is identical to running the GREB model with correction terms) and with a bare world. The latter is the GREB model with all switches OFF (radiative balance without an atmosphere and a dark surface). In comparison with the full GREB model, this illustrates how much all the climate processes affect the climate.

The GREB model without correction terms does capture the main features of the zonal mean climate, the seasonal cycle, the land-sea contrast and even smaller scale structures within continents or ocean basins (e.g. seasonal cycle structure within Asia or zonal temperature gradients within ocean basins). For most of the globe (<50° from the equator), the GREB model root-mean-squared error (RMSE) for the annual mean $T_{surf}$ is less than 10°C relative to the observed (see Fig. 4g). This is larger than for state-of-the-art CMIP-type climate models, which typically have an RMSE of about 2°C [Dommenget 2012]. In particular, the regions near the poles have high RMSE. It seems likely that the meridional heat transport is the main limitation in the GREB model, given the too warm tropical regions and the, in general, too cold polar regions and the too strong seasonal cycle in the polar regions in the GREB model without correction terms.

The GREB model performance can be put in perspective by illustrating how much the climate processes simulated in the GREB model contribute to the mean climate relative to the bare world simulation (see Fig. 4). The GREB RMSE to observed is about 20-30% of the RMSE of the bare world simulation (not shown), suggesting that the GREB model has a relative error of about 20-30% in the processes that it simulates or due to processes that it does not simulate (e.g. ocean heat transport).

**b. Mean climate deconstruction**

Understanding what is causing the mean observed climate with its regional and seasonal difference is often central for understanding climate variability and change. For instance, the seasonal cycle is often considered as a first guess estimate for climate sensitivity [Knutti et al. 2006]. In the following analysis, we will give a short overview on how the 10 processes of the MSCM experiments contribute to the mean climate and its seasonal cycle.

In Figures 5 and 6 the contribution of each of the 10 processes (except the atmosphere) to the annual mean climate (Fig. 5) and its seasonal cycle (Fig. 6) are shown. In each experiment, all processes are active, but the process of interest and the model correction terms are turned OFF. The results are compared against the complete GREB model without the model correction terms





(all processes active; expect model correction terms). For the hydrological we
will discuss some additional experiments in which the ice-albedo feedback is
turned OFF as well.
The Ice/Snow cover (Fig. 5a) has a strong cooling effect mostly at the high
latitudes in the cold season, which is due to the ice-albedo feedback. However, in
the warm season (not shown) the insulation effect of the sea ice actually leads to
warming, as the ocean cannot cool down as much during winter as it does
without sea ice.
Clouds (Fig. 5b) have a large net cooling effect globally due to the solar radiation
reflection effect dominating over the thermal radiation warming effect. It is also
interesting to note that the strongest cooling effect of cloud cover is over regions
with fairly little cloud cover (e.g. deserts and mountain regions). This is due to
the interaction with other climate feedbacks such as the water vapour feedback.
Previous studies on the cloud cover effect on the overall climate mostly focus on
the radiative forcings estimates, but to our best knowledge do not present the
overall change in surface temperature [e.g. Rossow and Zhang 1995].
The large ocean heat capacity slows down the seasonal cycle (Fig. 6c).
Subsequently, the seasons are more moderate than they would be without the
ocean transferring heat from warm to cold seasons. This is, in particular,
important in the mid and higher latitudes. The effect of the ocean heat capacity,
however, has also an annual mean warming effect (Fig. 5c). This is due to the
non-linear thermal radiation cooling. The non-linear black body negative
radiation feedback is stronger for warmer temperatures, which are not reached
in a moderated seasonal cycle with the larger ocean heat capacity.
The diffusion of heat reduces temperature extremes (Fig. 5d). It therefore warms
extremely cold regions (e.g. polar regions) and cools the hottest regions (e.g.
warm deserts). In global averages, this is mostly cancelled out. The advection of
heat has strong effects where the mean winds blow across strong temperature
gradients. This is mostly present in the Northern Hemisphere (Fig. 5e). The most
prominent feature is the strong warming of the northern European and Asian
continents in the cold season. In global average, warming and cooling mostly
cancel out.
The $CO_2$ concentration leads to global averages, warming of about 9 degrees (Fig.
5f). Even though it is the same $CO_2$ concentration everywhere, the warming effect
is different at different locations. This is discussed in more detail in DF11 and in
section 3c.
The input of water vapour into the atmosphere by the hydrological cycle leads to
a substantial amount of warming globally (Fig. 5g). However, we need to
consider that the experiment with switching OFF the hydrological cycle is the
only experiment in which we have a significant amount of global cooling (by
about -44°C). As a result, most of the earth is below freezing temperatures and
therefore has a much stronger ice-albedo feedback than in any other experiment.
This leads to a significant amplification of the response.
It is instructive to repeat the experiments with the ice-albedo feedback switched
OFF (see supplementary Fig. 1). In these experiments, all processes show a
reduced impact on the annual mean temperatures, but the hydrological cycle is
most strongly affected by it. The ice-albedo effect almost doubles the
hydrological cycle response, while for all other processes the effect is about a
10% to 40% increase. In the following discussions, we will therefore consider

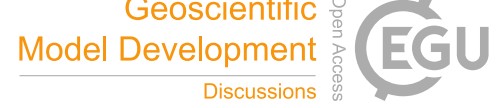

the hydrological cycle impact with and without ice-albedo feedback. In the
average of both response (Fig. 5g and SFig. 1g) the hydrological cycle has a global
mean impact of about +34°C with strongest amplitudes in the tropics. It is still
the strongest of all processes.
Similar to the oceans, it dampens the seasonal cycle (Fig. 6g), but with a much
weaker amplitude. The transport of water vapour away from warm and moist
regions (e.g. tropical oceans) to cold and dry regions (e.g. high latitudes and
continents) leads to additional warming in the regions that gain water vapour
and cooling to those that lose water vapour (Fig. 6h). The effect is similar in both
hemispheres. The transport of water vapour along the mean wind directions has
stronger effects on the Northern Hemisphere than on the Southern Hemisphere,
since the northern hemispheric mean winds have more of a meridional
component, which creates advection across water vapour gradients (Fig. 6i). This
effect is most pronounced in the cold seasons.
Most processes have a predominately zonal structure. We can therefore take a
closer look at the zonal mean climate and seasonal cycle of all processes to get a
good representation of the relative importance of each process, see Fig. 7. The
annual mean climate is most strongly influenced by the hydrological cycle (here
shown as the mean of the response with and without the ice-albedo feedback).
The cloud cover has an opposing cooling effect, but is weaker than the warming
effect of the hydrological cycle. The warming effect by the ocean's heat capacity
is similar in scale to that of the $CO_2$ concentration.
The seasonal cycle is damped most strongly by the ocean's heat capacity and by
the hydrological cycle. The later may seem unexpected, but is due to the effect
that the increased water vapour has a stronger warming effect in the cold
seasons, similarly to the greenhouse effect of $CO_2$ concentrations. In turn, the
ice/snow cover and cloud cover lead to an intensification of the seasonal cycle at
higher latitudes. Again, the later may seem unexpected, but is due to the
interaction with other climate feedbacks such as the water vapour feedback,
which also makes the climate more strongly respond to changes in cloud cover in
regions where there actually is very little cloud cover (e.g. deserts).
As an alternative way of understanding the role of the different process we can
build up the complete climate by introducing one process after the other, see
Figs. 8 and 9. We start with the bare earth (e.g. like our Moon) and then
introduce one process after the other. The order in which the processes are
introduced is mostly motivated by giving a good representation for each of the
10 processes. However, it can also be interpreted as a build up the Earth climate
in a somewhat historical way: We assume that initially the earth was a bare
planet and then the atmosphere, ocean, and all the other aspects were build up
over time.
The Bare Earth (all switches OFF) is a planet without atmosphere, ocean or ice. It
has an extremely strong seasonal cycle (Fig. 9a) and is much colder than our
current climate (Fig. 8a). It also has no regional structure other than meridional
temperature gradients. The combination of all climate processes will create most
of the regional and seasonal difference that make our current climate.
The atmospheric layer in the GREB model simulates two processes, if all other
processes are turned off: a turbulent sensible heat exchange with the surface and
thermal radiation due to residual trace gasses other than $CO_2$, water vapour or
clouds. However, as mentioned in the appendix A1 the log-function





approximation leads to negative emissivity if all greenhouse gasses ($CO_2$ and
water vapour) concentrations and cloud cover are zero. The negative emissivity
turns the atmospheric layer into a cooling effect, which dominates the impact of
the atmosphere in this experiment (Figs. 8b, c). This is a limitation of the GREB
model and the result of this experiment as such should be considered with
caution. In a more realistic experiment we can set the emissivity of the
atmosphere to zero or a very small value (0.01) to simulate the effect of the
atmosphere without $CO_2$, water vapour and cloud cover, see SFig. 2. Both
experiments have very similar warming effects in polar regions. Suggesting that
the sensible heat exchange warms the surface. The residual thermal radiation
effect from the emissivity of 0.01 has only a minor impact (SFig. 2f and g).
The warming effect of the $CO_2$ concentration is nearly uniform (Figs. 8d, e) and
without much of a seasonal cycle (Figs. 9d, e), if all other processes are turned
OFF. This accounts for a warming of about +9°C.
The oceans slow down the seasonal cycle by their large heat capacity (Figs. 9f, g).
The effective heat capacity of the oceans is proportional to the observed mixed
layer in the GREB model, which causes some small variations (differences from
the zonal means) as seen in the seasonal cycle of the oceans. Land points are not
affected, since no atmospheric transport exist (advection and diffusion turned
OFF). The different heat capacity between oceans and land already make a
significant element of the regional and seasonal climate differences (Figs. 8f, g).
Introducing turbulent diffusion of heat in the atmosphere now enables
interaction between points, which has the strongest effects along coastlines and
in higher latitudes (Figs. 8h, i). It reduces the land-sea contrast and has strong
effects over land with warming in winter and cooling in summer (Figs. 9h, i). The
extreme climates of the winter polar region are most strongly affected by the
turbulent heat exchange with lower latitudes. The turbulent heat exchange
makes the regional climate difference again a bit more realistic.
The advection of heat is strongly dependent on the temperature gradients along
the mean wind field directions. It provides substantial heating during the winter
season for Europe, Russia, and western North America (Figs. 8j, k, 9j, k). The
structure (differences from the zonal mean) created by this process is mostly
caused by the prescribed mean wind climatology. In particular, the milder
climate in Europe compared to northeast Asia on the same latitudes, are created
by wind blowing from the ocean onto land. The same is true for the differences
between the west and east coasts of the northern North America. The climate
regional and seasonal structures are now already quite realistic, but the overall
climate is much too cold. The ice/snow cover further cools the climate, in
particular, the polar regions (Figs. 8l, m). This difference illustrates that the ice-
albedo feedback is primarily leading to cooling in higher latitudes and mostly in
the winter season.
Introducing the hydrological cycle brings the most important greenhouse gas
into the atmosphere: water vapour. This has an enormous warming effect
globally (Figs. 8n, o) and a moderate reduction in the strength of the seasonal
cycle (Figs. 9n, o). The resulting modelled climate is now much too warm, but
introducing the cloud cover cools the climate substantially (Figs. 8p, q) and leads
to a fairly realistic climate.
The atmospheric transport (diffusion and advection) brings water vapour from
relative moist regions to relatively dry regions (Figs. 8r, s). This leads to




enhanced warming in the dry and cold regions (e.g. Sahara Desert or polar regions) by the water vapour thermal radiation (greenhouse) effect and cooling in the regions where it came from (e.g. tropical oceans). The heating effect is similar to the transport of heat and has also a strong seasonal cycle component.

**c. 2x$CO_2$ response deconstruction**

The doubling of the $CO_2$ concentrations leads to a distinct warming pattern with polar amplification, a land-sea contrast and significant seasonal differences in the warming rate. These structures in the warming pattern reflect the complex interactions between feedbacks in the climate system and regional difference in $CO_2$ forcing pattern. The MSCM 2x$CO_2$ response experiments are designed to help understand the interactions causing this distinct warming pattern. DF11 discussed many aspects of these experiments with focus on the land-sea contrast, the seasonal differences, and the polar amplification. We therefore will focus here only on some aspects that have not been previously discussed in DF11.

In the GREB model, we can turn OFF the atmospheric transport and therefore study the local interaction without any lateral interactions. Figure 10 shows three experiments in which the atmospheric transport and other processes (see Figure caption) are inactive. The three experiments highlight the regional difference in the $CO_2$ forcing pattern and in the two main feedbacks (water vapour and ice-albedo).

In the first experiment (Fig. 10a) without feedback processes, the local $T_{surf}$ response is approximately directly proportional to the local $CO_2$ forcing. The regional differences are caused by differences in the cloud cover and atmospheric humidity, since both influence the thermal radiation effect of $CO_2$ [DF11, Kiehl and Ramanathan 1982 and Cess et al. 1993]. This causes, on average, the land regions to see a stronger forcing than oceanic regions (see Fig. 10b). However, even over oceans we can see clear differences. For instance, the warm pool of the western tropical Pacific sees less $CO_2$ forcing than the eastern tropical Pacific.

The ice-albedo feedback is strongly localized and it is strongest over the mid-latitudes of the northern continents and at the sea ice edge of around Antarctica (Figs. 10c and d). The water vapour feedback is far more wide-spread and stronger (Figs. 10e and f). It is strongest in relatively warm and dry regions (e.g. subtropical oceans), but also shows some clear localized features, such as the strong Arabian or Mediterranean Seas warming.

**d. Scenarios**

The set of scenario experiments in the MSCM simulations allows us to study the response of the climate system to changes in the external boundary conditions in a number of different ways. In the following, we will briefly illustrate some results from these scenarios and organize the discussion by the different themes in scenario experiments.

The CMIP project has defined a number of standard $CO_2$ concentration projection simulations, that give different RCP scenarios for the future climate change, see Fig. 11a. The GREB model sensitivity in these scenarios is similar to those of the CMIP database [Forster et al. 2013].



Idealized $CO_2$ concentration scenarios help to understand the response to the $CO_2$
forcing. In Figure 11b, we show the global mean $T_{surf}$ response to different scaling
factors of $CO_2$ concentrations. To first order, we can see that the global mean $T_{surf}$
response follows a logarithmic $CO_2$ concentration (e.g. any doubling of the $CO_2$
concentration leads to the same global mean $T_{surf}$ response; compare 2x$CO_2$ with
4x$CO_2$ or with in Fig.11b) as suggested in other studies [Myhre et al. 1998].
However, this relationship does breakdown if we go to very low $CO_2$
concentrations (e.g. zero $CO_2$ concentration) illustrating that the log-function
approximation of the $CO_2$ forcing effect is only valid within a narrow range far
away from zero $CO_2$ concentration.
The transient response time to $CO_2$ forcing can be estimated from idealized $CO_2$
concentration changes, see Fig. 11c. The step-wise change in $CO_2$ concentration
illustrates the response time of the global climate. In the GREB model, it takes
about 10yrs to get 80% of the response to a $CO_2$ concentration change (see step-
function response, Fig. 11c). In turn, the response to a $CO_2$ concentration wave
time evolution is a lag of about 3yrs. The fast versus slow response also leads to
different warming patterns with strong land-sea contrasts (not shown), that are
largely similar to those found in previous studies [Held et al. 2010].
The regional aspects of the response to a $CO_2$ concentration can also be studied
by partially increasing the $CO_2$ concentration in different regions, see Fig. 12. The
warming response mostly follows the regions where we partially changed the
$CO_2$ concentration, but there are some interesting variations in this. The partial
increase in the $CO_2$ concentration over oceans has a stronger warming impact
than the partial increase in the $CO_2$ concentration over land for most Southern
Hemisphere land regions. In turn, the land forcing has little impact for the ocean
regions. The boreal winter forcing has stronger impact on the Southern
Hemisphere than boreal summer forcing, suggesting that the warm season
forcing is, in general, more important than the cold season forcing. The only
exception to this is the Tibet-plateau region.
A series of scenarios focus on the impact of solar forcing. In Figure 11d, we show
the response to an idealized 11yr solar cycle. The global mean $T_{surf}$ response is
two orders of magnitude smaller than the response to a doubling of the $CO_2$
concentration, reflecting the weak amplitude of this forcing. This result is largely
consistent with the response found in GCM simulations [Cubasch et al. 1997], but
does not consider possible more complicated amplification mechanisms [Meehl
et al. 2009]. A change in the solar constant of +27W/m$^2$ has a global $T_{surf}$
warming response similar to a doubling of the $CO_2$ concentration, but with a
slightly different warming pattern, see Fig. 13. The warming pattern of a solar
constant change has a stronger warming where incoming sun light is stronger
(e.g. tropics or summer season) and a weaker warming in region with less
incoming sun light (e.g. higher latitudes or winter season). This is in general
agreement with other modelling studies [Hansen et al. 1997].
On longer paleo time scales (>10,000yrs), changes in the orbital parameters
affect the incoming sun light. Figure 14 illustrates the response to a number of
orbital solar radiation changes. Incoming radiation (sunlight) typical of the ice
age (231kyrs ago) has less incoming sunlight in the Northern Hemispheric
summer. However, it has every little annual global mean changes (Fig. 14a) due
to increases in sunlight over other regions and seasons. The $T_{surf}$ response
pattern in the zonal mean at the different seasons is very similar to the solar

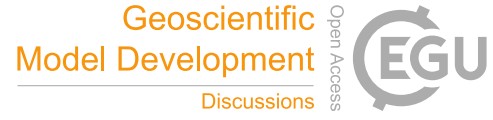



forcing, but the response is slightly more zonal and seasonal differences are less
dominant (Fig. 14b). The response is also amplified at higher latitudes. However,
in the global mean there is no significant global cooling as observed during ice
ages. If the solar forcing is combined with a reduction in the $CO_2$ concentration
(from 340ppm to 200ppm), we find a global mean cooling of -1.7$^o$C (Fig. 14c),
which is still much weaker than observed during ice ages, but is largely
consistent with previous studies of simulations of ice age conditions [Weaver et
al. 1998, Braconnot et al. 2007]. This is not unexpected since the GREB model
does not include an ice sheet model and, therefore, does not include glacier
growth feedbacks that would amplify ice age cycles.
A better understanding of the orbital solar radiation forcing can be gained by
analysing the response to idealized orbital parameter changes. We therefore
vary the Earth distance to the sun (radius), the earth axis tilt to the earth orbit
plane (obliquity) and shape of the earth orbit around the sun (eccentricity) over
a wider range, see Figs. 14 d-f. When the radius is changed by 10%, the Earth
climate becomes essentially uninhabitable, with either global mean temperature
above 30$^o$C (approx. summer mean temperature of the Sahara) or a completely
ice-covered snowball Earth. This suggests that the habitable zone of the Earth
radius is fairly small due to the positive feedbacks within the climate system
simulated in the GREB model (not considering long-term or more complex
atmospheric chemistry feedbacks) and largely consistent with previous studies
[Kasting et al. 1993].
When the obliquity is zero, the tropics become warmer and the polar regions
cool down further than today's climate, as they now receive very little sunlight
throughout the whole year. In the extreme case, when the obliquity is 90°, the
tropics become ice covered and cooler than the polar regions, which are now
warmer than the tropics today and ice free. The polar regions now have an
extreme seasonal cycle (not shown), with sunlight all day during summer and no
sunlight during winter.  Any eccentricity increase in amplitude would lead to a
warmer overall climate. Thus, a perfect circle orbit around the sun has, on
average, the coldest climate and all of the more extreme eccentricity (elliptic)
orbits have warmer climates. This suggests that the warming effect of the section
of the orbit that has a closer transit around the sun in an eccentricity orbit
relative to the perfect circle orbit overcompensates the cooling effect of the more
remote transit around the sun in the other half of the orbit relative to the perfect
circle orbit.

## 4. Summary and discussion

In this study, we introduced the MSCM database (version: MSCM-DB v1.0) for
research analysis with more than 1,300 experiments. It is based on model
simulations with the GREB model for studies of the processes that contribute to
the mean climate, the response to doubling of the $CO_2$ concentration, and
different scenarios with $CO_2$ or solar radiation forcings.  The GREB model is a
simple climate model that does not simulate internal weather variability,
circulation, or cloud cover changes. It provides a simple and fast null hypothesis
for the interactions in the climate system and its response to external forcings.
The GREB model without flux corrections simulates the mean observed climate
well and has an uncertainty of about 10˚C. The model has larger cold biases in

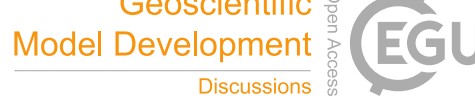



the polar regions indicating that the meridional heat transport is not strong
enough. Relative to a bare world without any climate processes the RMSE is
reduced to about 20-30% relative to observed. Thus, as a first guess, it can be
assumed that the GREB model simulations gives a 20-30% uncertainty in the
processes it simulates. Further, the GREB models emissivity function reaches
unphysical negative values when water vapour, $CO_2$ and cloud cover is set to
zero. This is a limitation of the log-function parametrization, that can potentially
be revised if a new parameterization is developed that considers these cases.
However, it is beyond the scope of this study to develop such a new
parameterization and it is left for future studies.
The MSCM experiments for the conceptual deconstruction of the observed mean
climate provide a good understanding of the processes that control the annual
mean climate and its seasonal cycle. The cloud cover, atmospheric water vapour,
and the ocean heat capacity are the most important processes that determine the
regional difference in the annual mean climate and its seasonal cycle. The
observed seasonal cycle is strongly damped not only by the ocean heat capacity,
but also by the water vapour feedback. In turn, ice-albedo and cloud cover
amplify the seasonal cycle in higher latitudes.
The conceptual deconstruction of the response to a doubling of the $CO_2$
concentration based on the MSCM experiments has mostly been discussed in
DF11, but some additional results shown here focused on the local forcing in
response without horizontal interaction. It has been shown here that the $CO_2$
forcing has a clear land-sea contrast, supporting the land-sea contrast in the $T_{surf}$
response. The water vapour feedback is wide-spread and most dominant over
the subtropical oceans, whereas the ice-albedo feedback is more localized over
Northern Hemispheric continents and around the sea ice border.
The series of scenario simulations with $CO_2$ and solar forcing provide many
useful experiments to understand different aspects of the climate response. The
RCP and idealized $CO_2$ forcing scenarios give good insights into the climate
sensitivity, regional differences, transient effects, and the role of $CO_2$ forcing at
different seasons or locations. The solar forcing experiments illustrate the subtle
differences in the warming pattern to a $CO_2$ forcing and the orbital solar forcing
illustrated elements of the climate response to long term, paleo, climate forcings.
In summary, the MSCM provides a wide range of experiments for understanding
the climate system and its response to external forcings. It builds a basis on
which conceptual ideas can be tested to a first-order and it provides a null
hypothesis for understanding complex climate interactions. Some of the
experiments presented here are similar to previously published simulations. In
general, the GREB model results agree well with the results of more complex
GCM simulations. It is beyond the scope of this study to discuss all aspects of the
experiments and their results. This will be left to future studies.
Future development of this MSCM database will continue and it is expected that
this database will grow. The development will go in several directions: the GREB
model performance in the processes that it currently simulates will be further
improved. In particular, the simulation of the hydrological cycle needs to be
improved to allow the use of the GREB model to study changes in precipitation.
Simulations of aspects of the large-scale atmospheric circulation, aerosols,
carbon cycle, or glaciers would further enhance the GREB model and would
provide a wider range of experiments to run for the MSCM database.





## 5. Code availability

The MSCM model code, including all required input files, to do all experiments described on the MSCM homepage and in this paper, can be downloaded as compressed tar archive from the MSCM homepage under

http://mscm.dkrz.de/download/mscm-web-code.tar.gz

or from the bitbucket repository under

https://bitbucket.org/tobiasbayr/mscm-web-code

The data for all the experiments of the MSCM can be accessed via the MSCM webpage interface (DOI: 10.4225/03/5a8cadac8db60). The mean deconstruction experiments file names have an 11 digits binary code that describe the 11 process switches combination: 1=ON and 0=OFF. The digit from left to right present the following processes:

1. Model corrections
2. Ice albedo
3. Cloud cover
4. Advection of water vapour
5. Diffusion of water vapour
6. Hydrologic cycle
7. Ocean
8. $CO_2$
9. Advection of heat
10. Diffusion of heat
11. Atmosphere

For example, the data file *greb.mean.decon.exp-10111111111.gad* is the experiment with all processes ON, but ice albedo is OFF. The 2x $CO_2$ response deconstruction experiments file names have a 10 digits binary code that describe the 10 process switches combination. The digit from left to right present the following processes:

1. Ocean heat uptake
2. Advection of water vapour
3. Diffusion of water vapour
4. Hydrologic cycle
5. ice albedo
6. Advection of heat
7. Diffusion of heat
8. Humidity (climatology)
9. Clouds (climatology)
10. Topography (Observed)

For example, the data file *response.exp-0111111111.2xCO2.gad* is the experiment with all processes ON, but ocean heat uptake is OFF. The individual experiments can be chosen from the webpage interface by selecting the desired switch





combinations. Alternatively, all experiments can be downloaded in a combined
tar-file from the webpage interface.

## 810     Acknowledgments

This study was supported by the ARC Centre of Excellence for Climate System
Science, Australian Research Council (grant CE110001028).

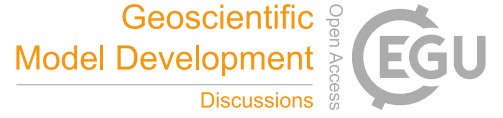



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



## Appendix A1: GREB model equations

The GREB model has four primary prognostic equations given below and all variable names are listed and explained in Table A1. The surface temperature, $T_{surf}$, tendencies:

$$\gamma_{surf} \frac{dT_{surf}}{dt} = F_{solar} + F_{thermal} + F_{latent} + F_{sense} + F_{ocean} + F_{correct} \quad [A1]$$

The atmospheric layer temperature, $T_{atmos}$, tendencies:

$$\gamma_{atmos} \frac{dT_{atmos}}{dt} = -F_{sense} + Fa_{thermal} + Q_{latent}$$
$$+\gamma_{atmos}(\kappa \cdot \nabla^2 T_{atmos} - \vec{u} \cdot \nabla T_{atmos}) \quad [A2]$$

The subsurface ocean temperature, $T_{ocean}$, tendencies:

$$\frac{dT_{ocean}}{dt} = \frac{1}{\Delta t} \Delta To_{entrain} - \frac{1}{\gamma_{ocean} - \gamma_{surf}} Fo_{sense} + Fo_{correct} \quad [A3]$$

The atmospheric specific humidity, $q_{air}$, tendencies:

$$\frac{dq_{air}}{dt} = \Delta q_{eva} + \Delta q_{precip} + \kappa \cdot \nabla^2 q_{air} - \vec{u} \cdot \nabla q_{air} + q_{correct} \quad [A4]$$

It should be noted here that heat transport is only within the atmospheric layer (eq. [A2]). Together with the moisture transport in eq. [A4] these transports are the only way in which grid points of the GREB model interact with each other in the horizontal directions.

The surface layer heat capacity, $\gamma_{surf}$, is constant over land points. For ocean points it follows the ocean mixed layer depth, $h_{mld}$, if $T_{surf}$ is above a temperature range near freezing. Within a range below freezing it is a linear increasing function of $T_{surf}$ and for $T_{surf}$ below this range $\gamma_{surf}$ the same as over land points. (see DF11).

The absorbed solar radiation, $F_{solar}$, is a function of the cloud cover, *CLD*, boundary condition and the surface albedo, $\alpha_{surf}$:

$$F_{solar} = (1 - \alpha_{clouds}) \cdot (1 - \alpha_{surf}) \cdot S_0 \cdot r \quad [A5]$$

with the atmospheric albedo, $\alpha_{clouds} = 0.35 \cdot CLD$. $\alpha_{surf}$ is a global constant if $T_{surf}$ is below or above a temperature range near freezing. Within this range it is a linear decreasing function of $T_{surf}$, (see DF11). The thermal radiation at the surface is

$$F_{thermal} = -\sigma T_{surf}^4 + \varepsilon_{atmos} \sigma T_{atmos-rad}^4 \quad [A6]$$

and the thermal radiation from the atmosphere is





$$Fa_{thermal} = \sigma T_{surf}^4 - 2\varepsilon_{atmos}\sigma T_{atmos-rad}^4 \qquad \text{[A7]}$$

The emissivity of the atmosphere, $\varepsilon_{atmos}$, is a function of the cloud cover, $CLD$,
the atmospheric water vapour, $viwv_{atmos}$, and the $CO_2$, $CO_2^{topo}$, concentration

$$\varepsilon_{atmos} = \frac{pe_8 - CLD}{pe_9} \cdot (\varepsilon_0 - pe_{10}) + pe_{10} \qquad \text{[A8]}$$

with

$$\varepsilon_0 = pe_4 \cdot \left[ pe_1 \cdot CO_2^{topo} + pe_2 \cdot viwv_{atmos} + pe_3 \right]$$
$$+ pe_5 \cdot \left[ pe_1 \cdot CO_2^{topo} + pe_3 \right] + pe_6 \cdot \left[ pe_2 \cdot viwv_{atmos} + pe_3 \right] + pe_7 \quad \text{[A9]}$$

The first three terms in the eq. [A9] represent different spectral bands in which
the thermal radiation of water vapour and the $CO_2$ are active. In the first term
both are active, in the second only $CO_2$ and in the third only water vapour. The
combined effect of eqs. [A8] and [A9] is that the sensitivity of the emissivity to
$CO_2$ is depending on the presents of cloud cover and water vapour.
It is important to note that this log-function parametrization of the emissivity is
an approximation developed in DF11 for 2x$CO_2$-concentration experiments.
While the parametrization may be a good approximation for a wide range of the
greenhouse gasses, it is likely to have limited skill in extreme variation of the
greenhouse gasses. For instance, if all greenhouse gasses ($CO_2$ and water vapour)
concentrations and cloud cover are zero then the emissivity of the atmospheric
layer in eq. [A9] becomes -0.26. This is not a physically meaningful value and
experiments in which all greenhouse gasses ($CO_2$ and water vapour) and cloud
cover are zero need to be analysed with caution. The analysis section will discuss
these limitations in these experiments.

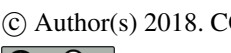



## Tables

**Table 1**: Processes (switches) controlled in the sensitivity experiment for the
mean climate deconstruction. Indentation in the left column indicates processes
switches are dependent on the switches above being ON.

| Mean Climate Deconstruction | |
|---|---|
| Name | Description |
| Ice-albedo | controls surface albedo ($\alpha_{surf}$) and heat capacity ($\gamma_{surf}$) at sea ice points as function of $T_{surf}$ |
| Clouds | controls cloud cover climatology. OFF equals no clouds. |
| Oceans | controls $F_{ocean}$ term in eq. [A1] and the heat capacity ($\gamma_{surf}$) off all ocean points. OFF equals no $F_{ocean}$ and as $\gamma_{surf}$ over land. |
| Atmosphere | controls sensible heat flux ($F_{sense}$) and the downward atmospheric thermal radiation term in eq. [A6]. |
|    Diffusion of Heat | controls diffusion of heat |
|    Advection of Heat | controls advection of heat |
|    $CO_2$ | controls $CO_2$ concentration |
|    Hydrological cycle | controls atmospheric humidity. OFF equals zero humidity |
|       Diffusion of water vapour | controls diffusion of water vapour |
|       Advection of water vapour | controls advection of water vapour |
| Model Corrections | controls model flux correction terms |





**Table 2**: Processes (switches) controlled in the sensitivity experiment for the
2xCO$_2$ response deconstruction. Indentation in the left column indicates
processes switches are dependent on the switches above being ON.

| 2xCO$_2$ Response Deconstruction | |
|---|---|
| Boundary Conditions | |
| Name | Description |
| Topography (Observed) | controls topography effect on thermal radiation. OFF equals all land point on sea level. |
| Clouds (climatology) | controls cloud cover climatology. OFF equals 0.7 cloud cover everywhere. |
| Humidity (climatology) | controls the humidity constraint. OFF equals a control humidity 0.0052 [kg/kg] everywhere. Humidity can still respond to forcings. |
| Feedbacks/Processes | |
| Diffusion of Heat | controls diffusion of heat |
| Advection of Heat | controls advection of heat |
| Ice-albedo | controls surface albedo ($\alpha_{surf}$) and heat capacity ($\gamma_{surf}$) at sea ice points as function of $T_{surf}$ |
| Ocean heat uptake | controls $F_{ocean}$ term in eq. [A1] and the heat capacity ($\gamma_{surf}$) off all ocean points. OFF equals no $F_{ocean}$ and $\gamma_{surf}$ of a 50m water column. |
| Hydrological cycle | controls atmospheric humidity. OFF equals zero humidity |
| Diffusion of water vapour | controls diffusion of water vapour |
| Advection of water vapour | controls advection of water vapour |




**Table 3**: List of scenario experiments.

| Name | length | Description |
|---|---|---|
| RCP CO$_2$-scenarios | | |
| Historical | 1850-2000 | CO$_2$-concentration following the historical scenario |
| RCP8.5 | 2001-2100 | CO$_2$-concentration following the RCP8.5 scenario |
| RCP6 | 2001-2100 | CO$_2$-concentration following the RCP6 scenario |
| RCP4 | 2001-2100 | CO$_2$-concentration following the RCP4 scenario |
| RCP3PD | 2001-2100 | CO$_2$-concentration following the RCP3PD scenario |
| A1B | 2001-2100 | CO$_2$-concentration following the A1B scenario |
| Idealized CO$_2$ concentrations | | |
| Zero-CO$_2$ | 100yrs | zero CO$_2$ concentrations |
| 0.5xCO$_2$ | 50yrs | 140ppm CO$_2$ concentrations |
| 2xCO$_2$ | 50yrs | 560ppm CO$_2$ concentrations |
| 4xCO$_2$ | 100yrs | 1120ppm CO$_2$ concentrations |
| 10xCO$_2$ | 100yrs | 2800ppm CO$_2$ concentrations |
| Partial CO$_2$ concentrations | | |
| CO$_2$-N-hemis | 50yrs | 2xCO$_2$ only in the northern hemisphere |
| CO$_2$-S-hemis | 50yrs | 2xCO$_2$ only in the southern hemisphere |
| CO$_2$-tropics | 50yrs | 2xCO$_2$ only between 30$^o$S and 30$^o$N |
| CO$_2$-extra-tropics | 50yrs | 2xCO$_2$ only poleward of 30$^o$ |
| CO$_2$-oceans | 50yrs | 2xCO$_2$ only over ice-free ocean points |
| CO$_2$-land | 50yrs | 2xCO$_2$ only over land and sea ice points |
| CO$_2$-winter | 50yrs | 2xCO$_2$ only in the month Oct. to Mar. |
| CO$_2$-summer | 50yrs | 2xCO$_2$ only in the month Apr. to Sep. |
| Solar radiation | | |
| solar+27W/m$^2$ | 50yrs | solar constant increased by +27W/m$^2$ |
| 11yrs-solar | 50yrs | solar idealized solar constant 11yrs cycle |
| Orbital parameter | | |
| Solar-231Kyr | 100yrs | incoming solar radiation according to orbital parameters 231Kyrs ago. |
| Solar-231Kyr-200ppm | 100yrs | as Solar-231Kyr, but with CO$_2$ concentrations decreased from 280ppm to 200ppm. |
| Orbit-radius | 40steps | equilibrium response to different Earth orbit radius from 0.8AU to 1.2AU. |
| Obliquity | 45steps | equilibrium response to different Earth axis tilt from -25$^o$ to 90$^o$ |
| Eccentricity | 60steps | equilibrium response to different Earth orbit eccentricity from 0.3 to 0.3 |





**Table A1**: Variables of the GREB model equations.

| Variable | Dimensions | Description |
|---|---|---|
| $T_{surf}$ | x, y, t | surface temperature |
| $T_{atmos}$ | x, y, t | atmospheric temperature |
| $T_{ocean}$ | x, y, t | subsurface ocean temperature |
| $q_{air}$ | x, y, t | atmospheric humidity |
| $\gamma_{surf}$ | x, y, t | heat capacity of the surface layer |
| $\gamma_{atmos}$ | x, y, t | heat capacity of the atmosphere |
| $\gamma_{ocean}$ | x, y, t | heat capacity of the subsurface ocean |
| $F_{solar}$ | x, y, t | solar radiation absorbed at the surface |
| $F_{thermal}$ | x, y, t | thermal radiation into the surface |
| $Fa_{thermal}$ | x, y, t | thermal radiation into the atmospheric |
| $F_{latent}$ | x, y, t | latent heat flux into the surface |
| $Q_{latent}$ | x, y, t | latent heat flux into the atmospheric |
| $F_{sense}$ | x, y, t | sensible heat flux from the atmosphere into the surface |
| $Fo_{sense}$ | x, y, t | sensible heat flux from the subsurface ocean into the surface layer |
| $F_{ocean}$ | x, y, t | sensible heat flux from the subsurface ocean |
| $F_{correct}$ | x, y, t | heat flux corrections for the surface |
| $Fo_{correct}$ | x, y, t | heat flux corrections for the subsurface ocean |
| $q_{correct}$ | x, y, t | mass flux corrections for the atmospheric humidity |
| $\Delta To_{entrain}$ | x, y, t | subsurface ocean temperature tendencies by entrainment |
| $\Delta q_{eva}$ | x, y, t | mass flux for the atmospheric humidity by evaporation |
| $\Delta q_{precip}$ | x, y, t | mass flux for the atmospheric humidity by precipitation |
| $\alpha_{surf}$ | x, y, t | albedo of the surface layer |
| $\varepsilon_{atmos}$ | x, y, t | emissivity of the atmosphere |
| $T_{atmos-rad}$ | x, y, t | atmospheric radiation temperature |
| $viwv_{atmos}$ | x, y, t | atmospheric column water vapour mass |
| $\kappa$ | constant | isotropic diffusion coefficient |
| $pe_i$ | constant | empirical emissivity function parameters |
| $\vec{u}$ | x, y, $t_j$ | horizontal wind field |
| $\alpha_{clouds}$ | x, y, $t_j$ | albedo of the atmosphere |
| $h_{mld}$ | x, y, $t_j$ | Ocean mixed layer depth |
| r | y, $t_j$ | fraction of incoming sunlight (24hrs average) |
| $CO_2^{topo}$ | x, y | $CO_2$ concentration scaled by topographic elevation |
| $S_0$ | constant | solar constant |
| $\sigma$ | constant | Stefan-Bolzman constant |
| $t_j$ | - | day within the annual calendar |
| $\Delta t$ | constant | model integration time step |
| $\sigma$ | constant | Stefan-Boltzmann constant |




## Figures

**Figure 1.** MSCM interface running the deconstruction of the mean climate experiments. The experiment A, on the left, has all processes turned ON and experiment B, on right, has all turned OFF. The $T_{surf}$ of Experiment A is shown in the upper left map, Exp. B in the upper right and the difference between both in the lower map. The example shows the values for the October mean.

**Figure 2.** MSCM interface running the deconstruction of the response to a doubling of the $CO_2$ concentration experiments. The experiment A, on the left, has all processes turned ON and experiment B, on right, has all turned OFF. The $T_{surf}$ response of Experiment A is shown in the upper left map, Exp. B in the upper right and the difference between both in the lower map. The example shows the annual mean values after 28yrs.

**Figure 3.** Examples of the MSCM scenario interface. (a) presenting a single scenario (here RCP 8.5 $CO_2$ forcing) and (b) the comparison of two different scenarios (here a $CO_2$ forcing is compared against a change in the solar constant by +27W/m$^2$).

**Figure 4.** $T_{surf}$ annual mean (upper row) and seasonal cycle (half the difference between mean of July to September minus January to March; middle row) for three different experiments: GREB with all processes turned OFF (Bare Earth), all processes on (observed) and only the correction term OFF (GREB). The zonal mean of the annual mean (g) and seasonal cycle (h) of the three experiments in comparison with the zonal mean RMSE of the GREB model without correction terms relative to observed.

**Figure 5.** Changes in the annual mean $T_{surf}$ in the GREB model simulations with different processes turned OFF as described in section 2a relative to the complete GREB model without model correction terms: (a) Ice/Snow, (b) clouds, (c) oceans, (d) heat advection, (e) heat diffusion, (f) $CO_2$ concentration, (g) hydrological cycle, (h) diffusion of water vapour and (i) advection of water vapour. Global mean differences are shown in the headings. Differences are for the control minus the sensitivity experiment (positive indicates the control experiment is warmer). All values are in ᵒC. In some panels, the values are scaled for better comparison: (b), (c) and (f) by a factor of 2, (a), (d) and (e) by a factor of 3, and (h) and (i) by a factor of 6.

**Figure 6.** As in Fig. 5, but for the seasonal cycle. The mean seasonal cycle is defined by the difference between the month [JAS] - [JFM] divided by two. Positive values on the North hemisphere indicate stronger seasonal cycle in the sensitivity experiments than in the full GREB model. Vice versa for the Southern Hemisphere. Global root mean square differences are shown in the headings. All values are in ᵒC. In some panels, the values are scaled for better comparison: (b), (d) and (e) by a factor of 2, and (h) and (i) by a





factor of 10. (g) is the mean for the hydrological cycle experiments with
and without the ice-albedo process active.

**Figure 7.** Zonal mean values of the annual mean (a) and seasonal cycle
differences (b) for the experiments as shown in Figs. 5 and 6. g) The mean
for the hydrological cycle is for the experiments with and without the ice-
albedo process active.

**Figure 8.** Conceptual build-up of the annual mean climate: staring with all
processes turned OFF (a) and then adding more processes in each row:
(b) atmosphere, (d) $CO_2$, (f) oceans, (h) heat diffusion, (j) heat advection,
(l) hydrological cycle, (n) ice-albedo, (p) clouds and (r) water vapour
transport. The panels on the right column show the difference of the left
panel to the previous row left panel. Global mean values are shown in the
heading. All values are in ᵒC. In some panels in the right column the values
are scaled for better comparison: (e), (g) and (q) by a factor of 2, (i) by a
factor of 3 and (k), (o) and (s) by a factor of 4. For details see on the
experiments see section 2a.

**Figure 9.** As in Fig. 8, but conceptual build-up of the seasonal cycle. The
seasonal cycle is defined by the difference between the month [JAS] -
[JFM] divided by two. Global mean absolute values are shown in the
heading. In some panels in the right column the values are scaled for
better comparison: (c), (i), (m) and (o) by a factor of 2, (k), (q) and (s) by
a factor of 5 and for (e) by a factor of 30.

**Figure 10.**      Local $T_{surf}$ response to doubling of the $CO_2$ concentration in
experiments without atmospheric transport (each point on the maps is
independent of the others). (a) GREB with topography, humidity and
cloud processes and all other processes OFF. (b) Difference of (a) to GREB
with topography and all other processes OFF scaled by a factor of 10. (c)
GREB model as in (a), but with ice-albedo process ON. (d) Difference of
(c)-(a) scaled by a factor of 2. (e) GREB model as in (a), but with
hydrological cycle process ON. (f) Difference of (e)-(a) scaled by a factor
of 2. For details see on the experiments see section 2b.

**Figure 11.**      Global mean $T_{surf}$ response to idealized forcing scenarios:
(a) different RCP $CO_2$ forcing scenarios. (b) Scaled $CO_2$ concentrations. (c)
idealized $CO_2$ concentration time evolutions (dotted lines) and the
respective $T_{surf}$ responses (solid lines of the same colour). (d) idealized
11yrs solar cycle. List of experiments is given in Table 3.

**Figure 12.**      $T_{surf}$ response to partial doubling of the $CO_2$ concentration
in: Northern (a) and Southern (b) hemisphere, tropics (d) and extra-
tropics (e), oceans (g) and land (h), and in boreal winter (j) and summer
(k) . The right column panels show the difference between the two panels
two the left in the same row.





**Figure 13.**  $T_{surf}$ response to changes in the solar constant by +27W/m$^2$ (middle column) versus a doubling of the $CO_2$ concentration (left column) for the annual mean (upper) and the seasonal cycle (lower). The seasonal cycle is defined by the difference between the month [JAS] - [JFM] divided by two. The right column panels show the difference between the two panels two the left in the same row scaled by 4 (c) and 3 (f).

**Figure 14.**  Orbital parameter forcings and $T_{surf}$ responses: (a) incoming solar radiation changes in the Solar-231Kyr experiment relative to the control GREB model. $T_{surf}$ response in Solar-231Kyr (b) and Solar-231Kyr-200ppm (c) relative to the control GREB model. Annual mean $T_{surf}$ in Orbit-radius (d), Obliquity (e) and Eccentricity (f). The solid vertical line in (d)-(f) marks the control (today) GREB model.



Figure 1

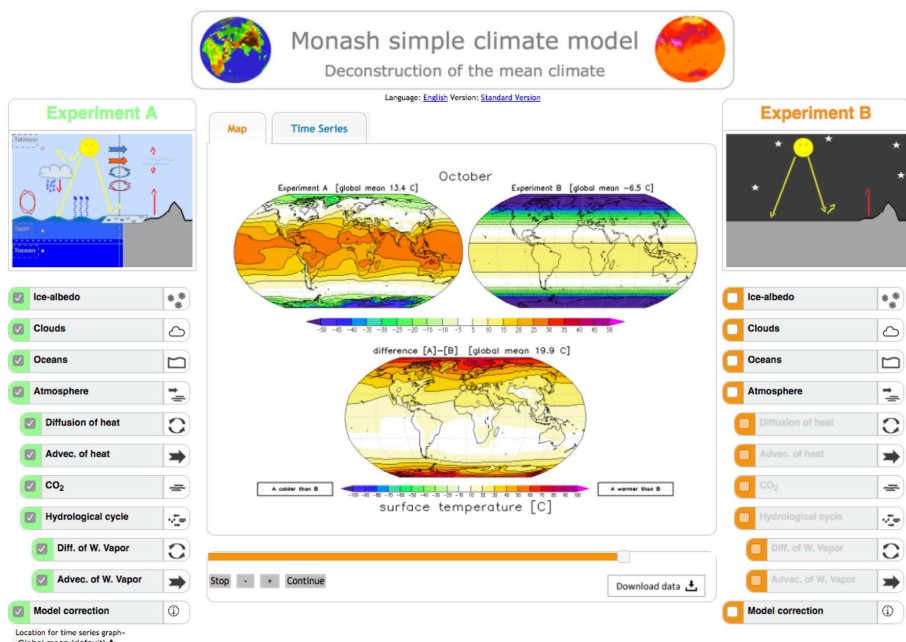

Figure 1: MSCM interface running the deconstruction of the mean climate experiments. The experiment A, on the left, has all processes turned ON and experiment B, on right, has all turned OFF. The $T_{surf}$ of Experiment A is shown in the upper left map, Exp. B in the upper right and the difference between both in the lower map. The example shows the values for the October mean.



Figure 2

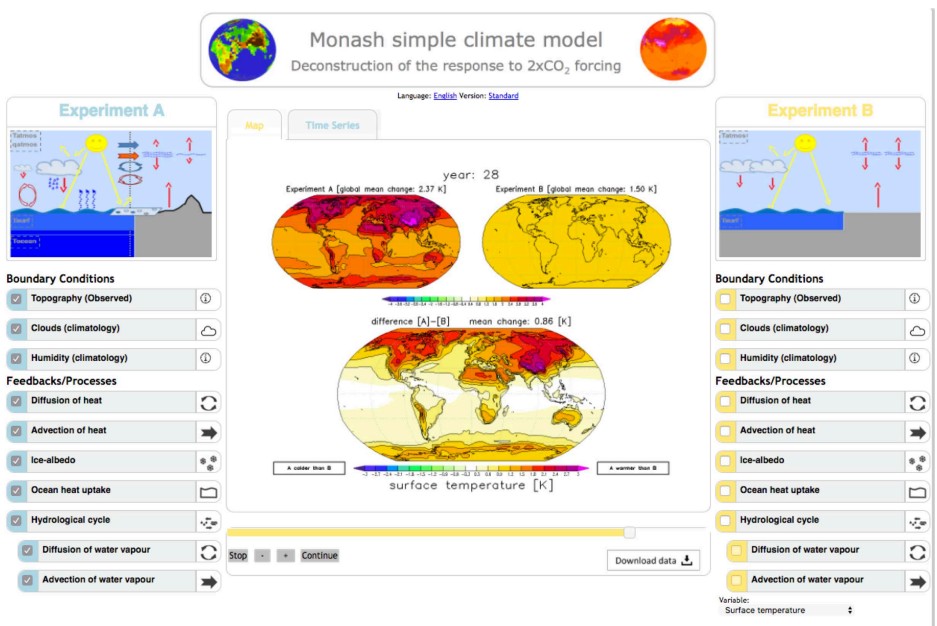

Figure 2: MSCM interface running the deconstruction of the response to a doubling of the $CO_2$ concentration experiments. The experiment A, on the left, has all processes turned ON and experiment B, on right, has all turned OFF. The $T_{surf}$ response of Experiment A is shown in the upper left map, Exp. B in the upper right and the difference between both in the lower map. The example shows the annual mean values after 28yrs.



Figure 3

(a)

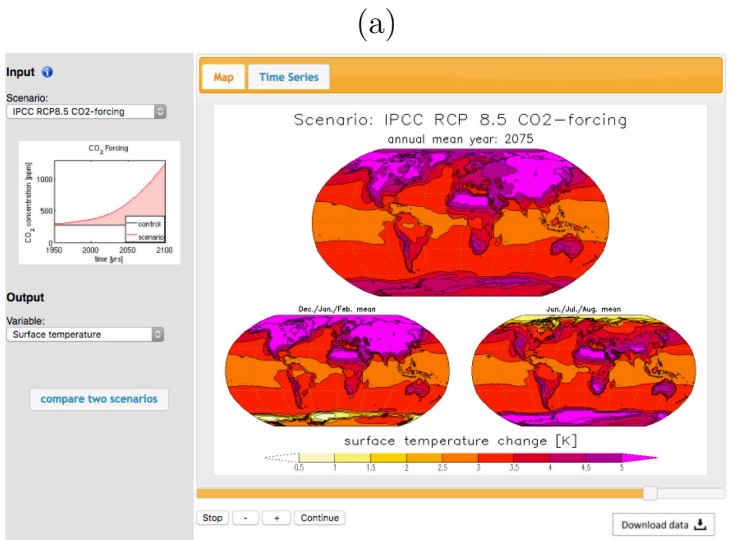

(b)

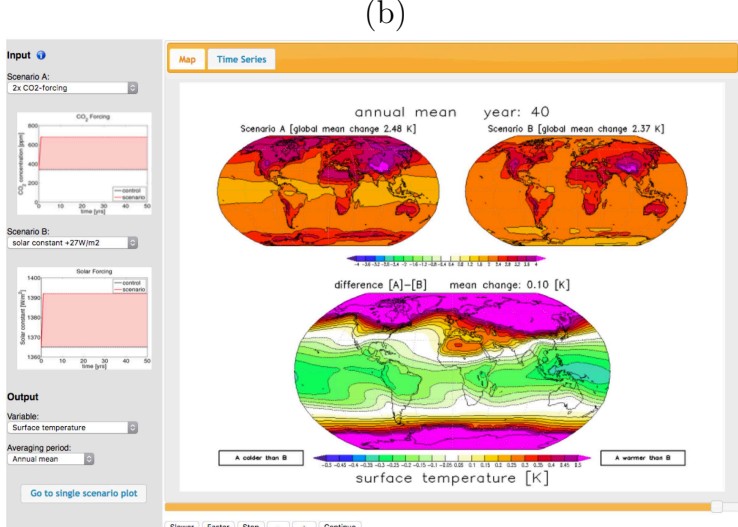

Figure 3: Examples of the MSCM scenario interface. (a) presenting a single scenario (here RCP 8.5 $CO_2$ forcing) and (b) the comparison of two different scenarios (here a $CO_2$ forcing is compared against a change in the solar constant by $+27W/m^2$).





Figure 4



Figure 4: Tsurf annual mean (upper row) and seasonal cycle (half the difference between mean of July to September minus January to March; middle row) for three different experiments: GREB with all processes turned OFF (Bare Earth), all processes on (observed) and only the correction term OFF (GREB). The zonal mean of the annual mean (g) and seasonal cycle (h) of the three experiments in comparison with the zonal mean RMSE of the GREB model without correction terms relative to observed.





Figure 5

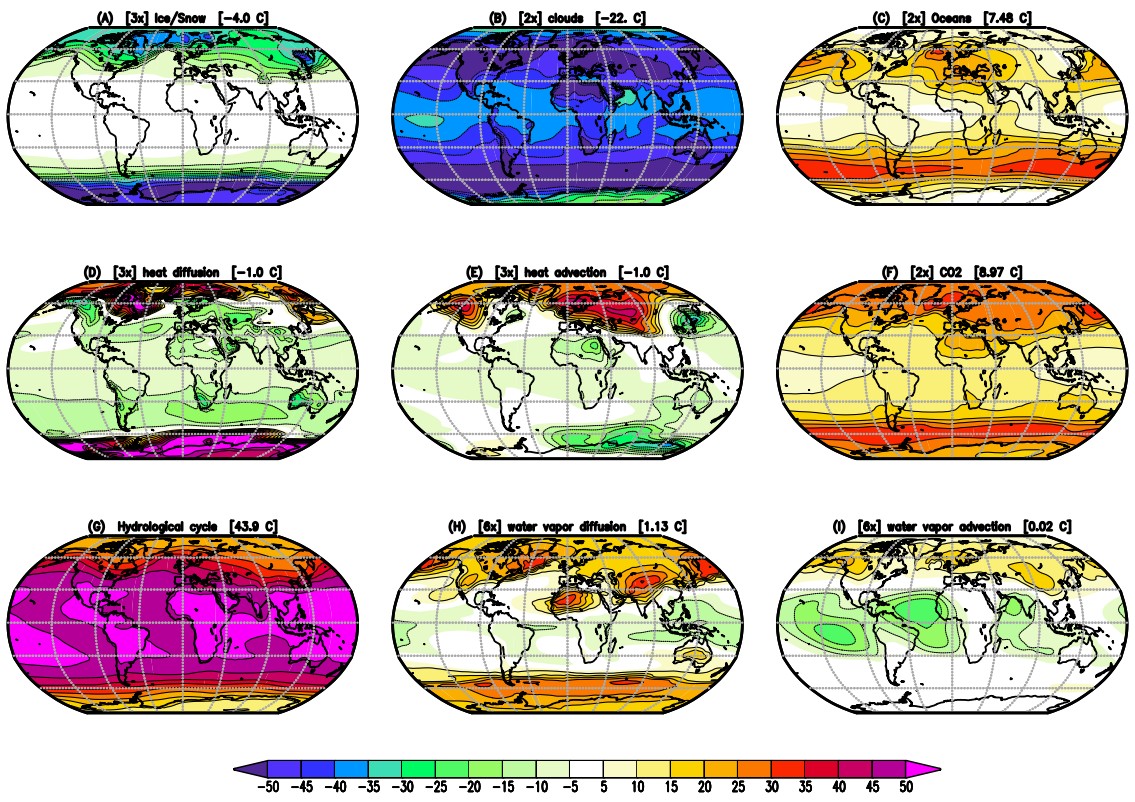

Figure 5: Changes in the annual mean $T_{surf}$ in the GREB model simulations with different processes turned OFF as described in section 2a relative to the complete GREB model without model correction terms: (a) Ice/Snow, (b) clouds, (c) oceans, (d) heat advection, (e) heat diffusion, (f) $CO_2$ concentration, (g) hydrological cycle, (h) diffusion of water vapour and (i) advection of water vapour. Global mean differences are shown in the headings. Differences are for the control minus the sensitivity experiment (positive indicates the control experiment is warmer). All values are in $^oC$. In some panels, the values are scaled for better comparison: (b), (c) and (f) by a factor of 2, (a), (d) and (e) by a factor of 3, and (h) and (i) by a factor of 6.





Figure 6

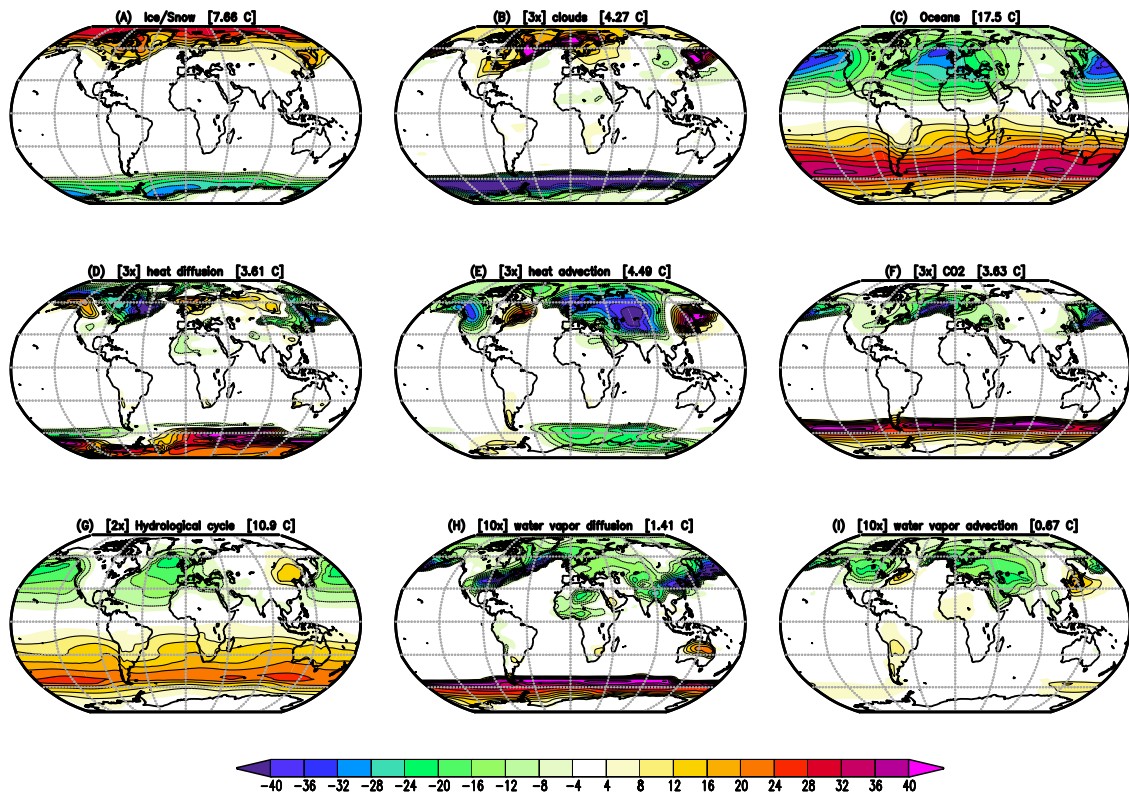

Figure 6: As in Fig. 5, but for the seasonal cycle. The mean seasonal cycle is defined by the difference between the month [JAS] - [JFM] divided by two. Positive values on the North hemisphere indicate stronger seasonal cycle in the sensitivity experiments than in the full GREB model. Vice versa for the Southern Hemisphere. Global root mean square differences are shown in the headings. All values are in $^oC$. In some panels, the values are scaled for better comparison: (b), (d) and (e) by a factor of 2, and (h) and (i) by a factor of 10. (g) is the mean for the hydrological cycle experiments with and without the ice-albedo process active.




Figure 7

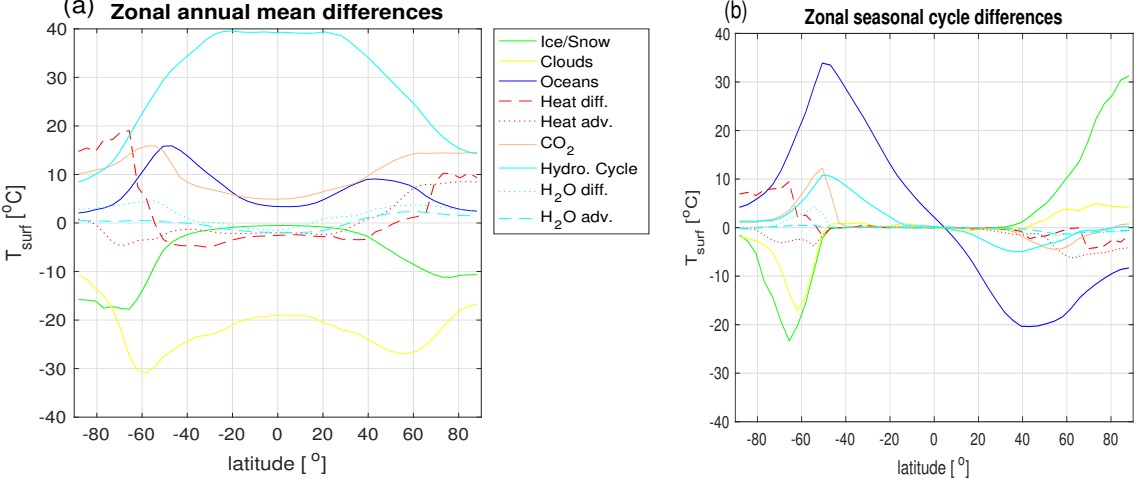

Figure 7: Zonal mean values of the annual mean (a) and seasonal cycle differences (b) for
the experiments as shown in Figs. 5 and 6. g) The mean for the hydrological cycle is for the
experiments with and without the ice-albedo process active.



Figure 8 part 1

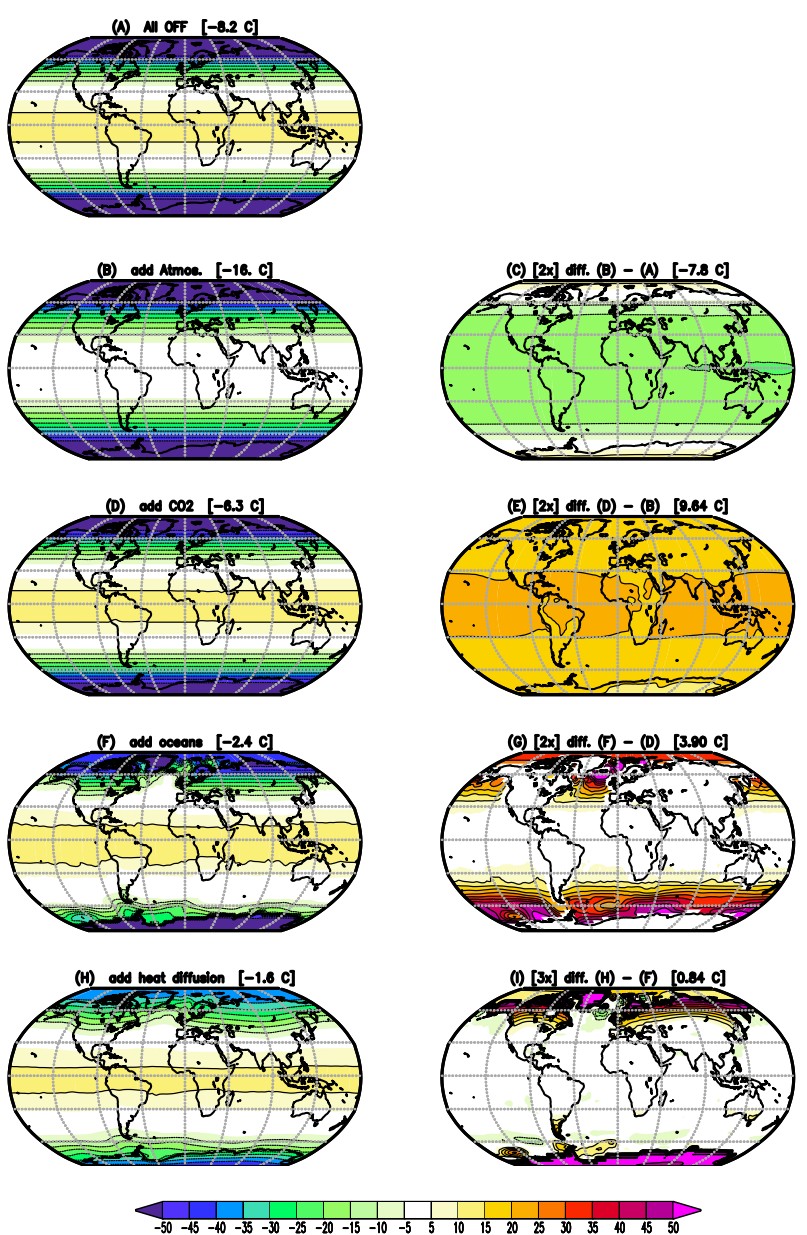

Figure 8: Conceptual build-up of the annual mean climate: staring with all processes turned OFF (a) and then adding more processes in each row: (b) atmosphere, (d) CO2, (f) oceans, (h) heat diffusion, (j) heat advection, (l) ice-albedo, (n) hydrological cycle, (p) clouds and (r) water vapour transport. The panels on the right column show the difference of the left panel to the previous row left panel. Global mean values are shown in the heading. All values are in oC. In some panels in the right column the values are scaled for better comparison: (e), (g) and (q) by a factor of 2, (i) and (m) by a factor of 3 and (c), (k) and (s) by a factor of 4. For details see on the experiments see section 2a.



Figure 8 part 2

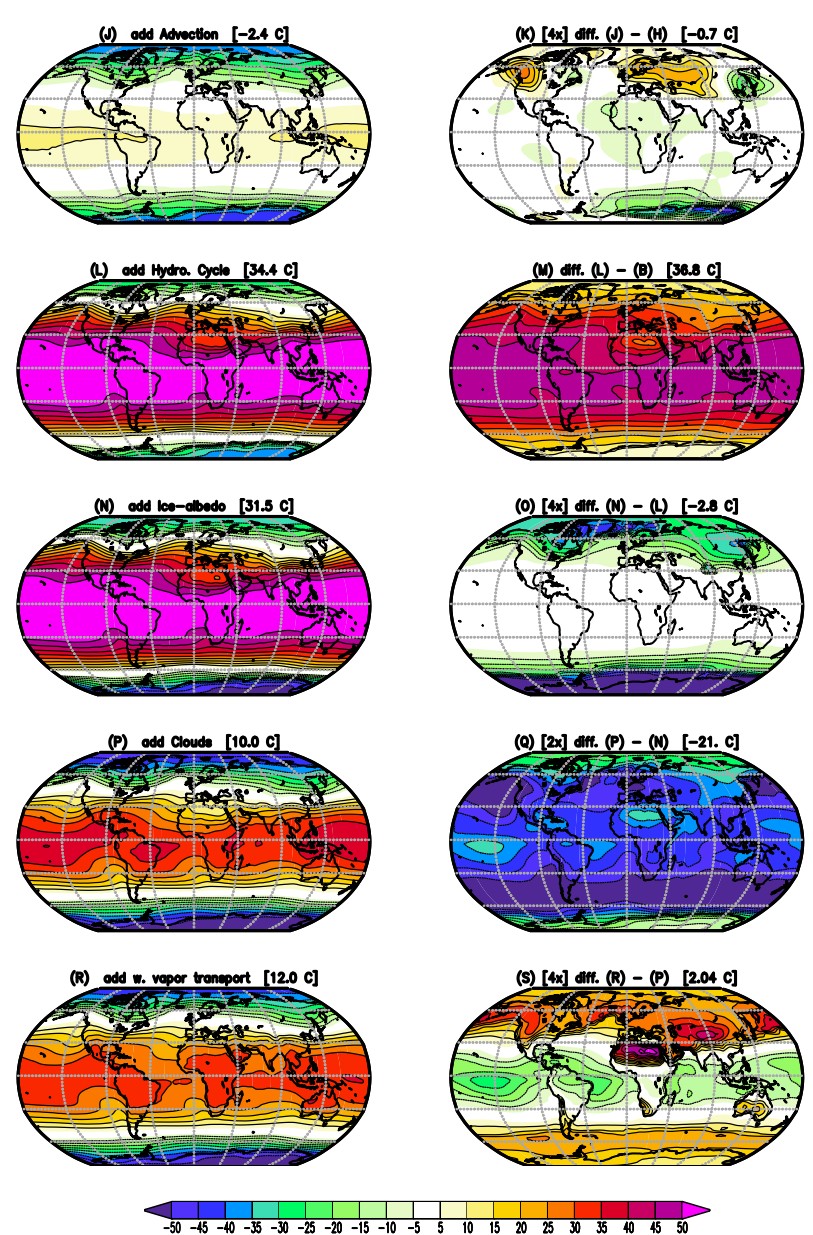





Figure 9 part 1

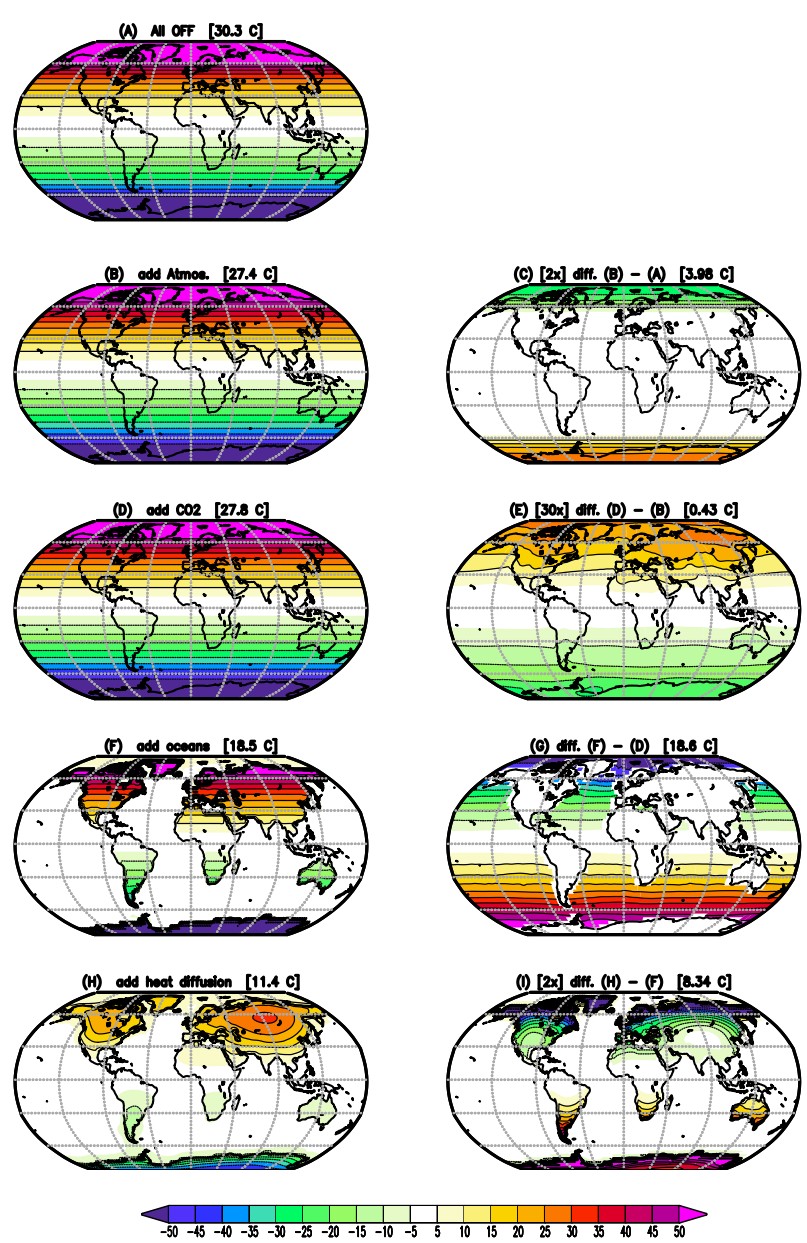

Figure 9: As in Fig. 8, but conceptual build-up of the seasonal cycle. The seasonal cycle is defined by the difference between the month [JAS] - [JFM] divided by two. Global mean absolute values are shown in the heading. In some panels in the right column the values are scaled for better comparison: (c) and (o) by a factor of 2, (i), (k), (q) and (s) by a factor of 5 and for (e) by a factor of 30.





Figure 9 part 2

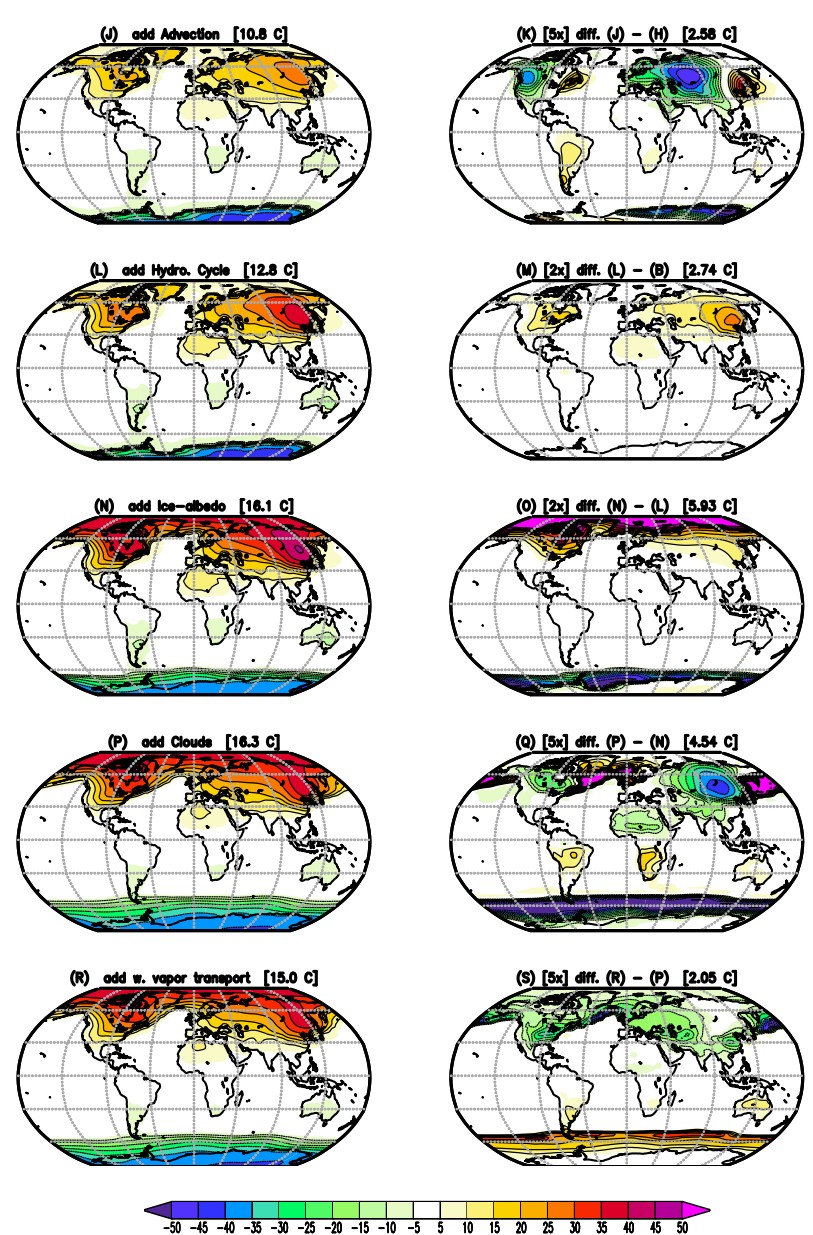



Figure10

Figure 10: Local $T_{surf}$ response to doubling of the $CO_2$ concentration in experiments without atmospheric transport (each point on the maps is independent of the others). (a) GREB with topography, humidity and cloud processes and all other processes OFF. (b) difference of (a) to GREB with topography and all other processes OFF scaled by a factor of 10. (c) GREB model as in (a), but with ice-albedo process ON. (d) difference of (c)-(a) scaled by a factor of 2. (e) GREB model as in (a), but with hydrological cycle process ON. (f) difference of (e)-(a) scaled by a factor of 2. For details see on the experiments see section 2b.




Figure 11

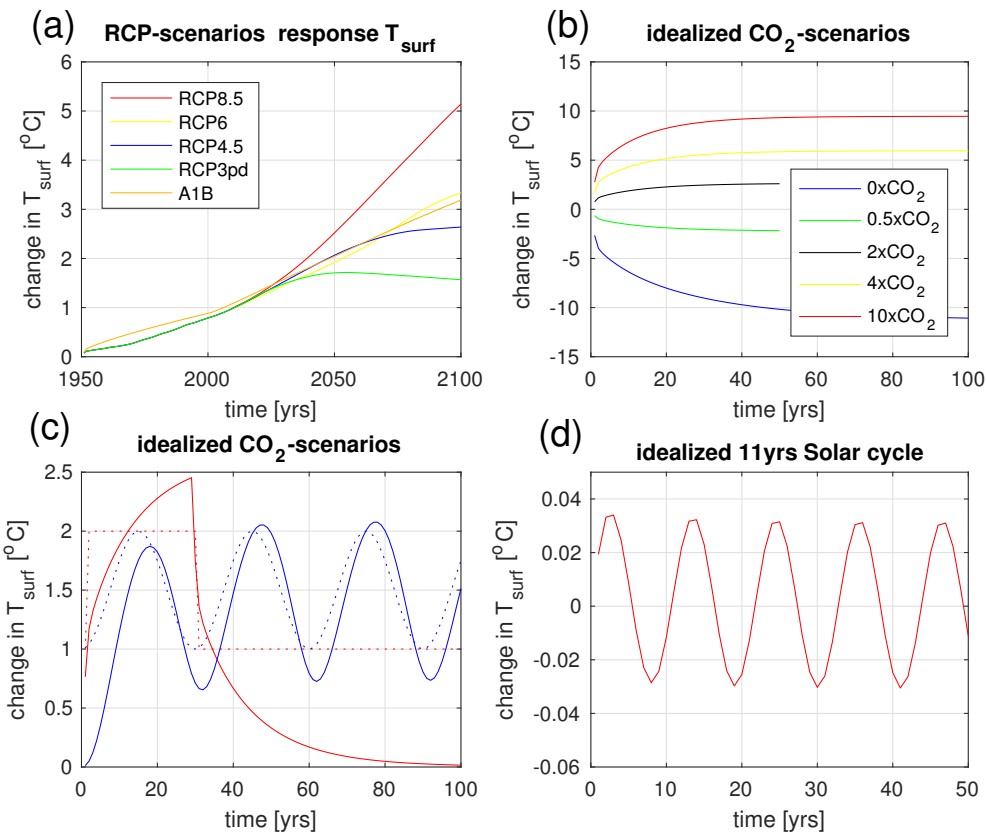

Figure 11: Global mean Tsurf response to idealized forcing scenarios: (a) different RCP $CO_2$ forcing scenarios. (b) Scaled $CO_2$ concentrations. (c) idealized $CO_2$ concentration time evolutions (dotted lines) and the respective Tsurf responses (solid lines of the same colour). (d) idealized 11yrs solar cycle. List of experiments is given in Table 3.



Figure 12

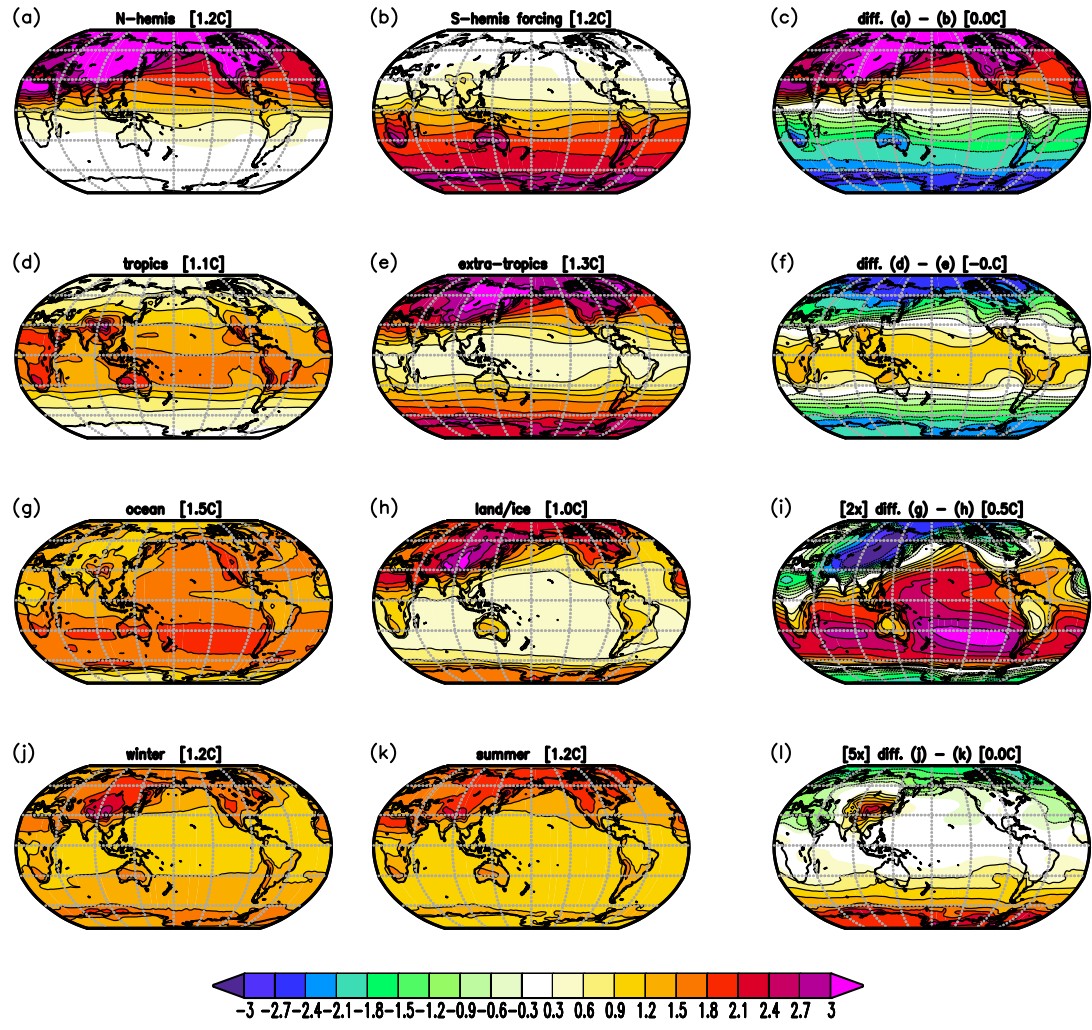

Figure 12: Tsurf response to partial doubling of the CO2 concentration in: Northern (a) and Southern (b) hemisphere, tropics (d) and extra-tropics (e), oceans (g) and land (h), and in boreal winter (j) and summer (k) . The right column panels show the difference between the two panels two the left in the same row.





Figure 13

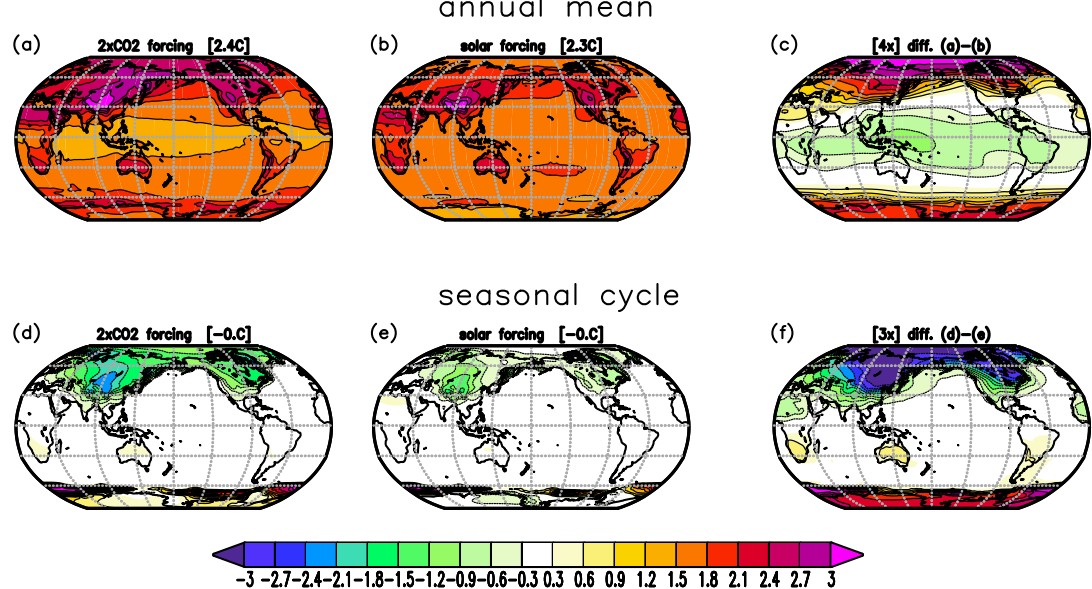

Figure 13: Tsurf response to changes in the solar constant by $+27W/m^2$ (middle column) versus a doubling of the $CO_2$ concentration (left column) for the annual mean (upper) and the seasonal cycle (lower). The seasonal cycle is defined by the difference between the month [JAS] - [JFM] divided by two. The right column panels show the difference between the two panels two the left in the same row scaled by 4 (c) and 3 (f).



Figure 14

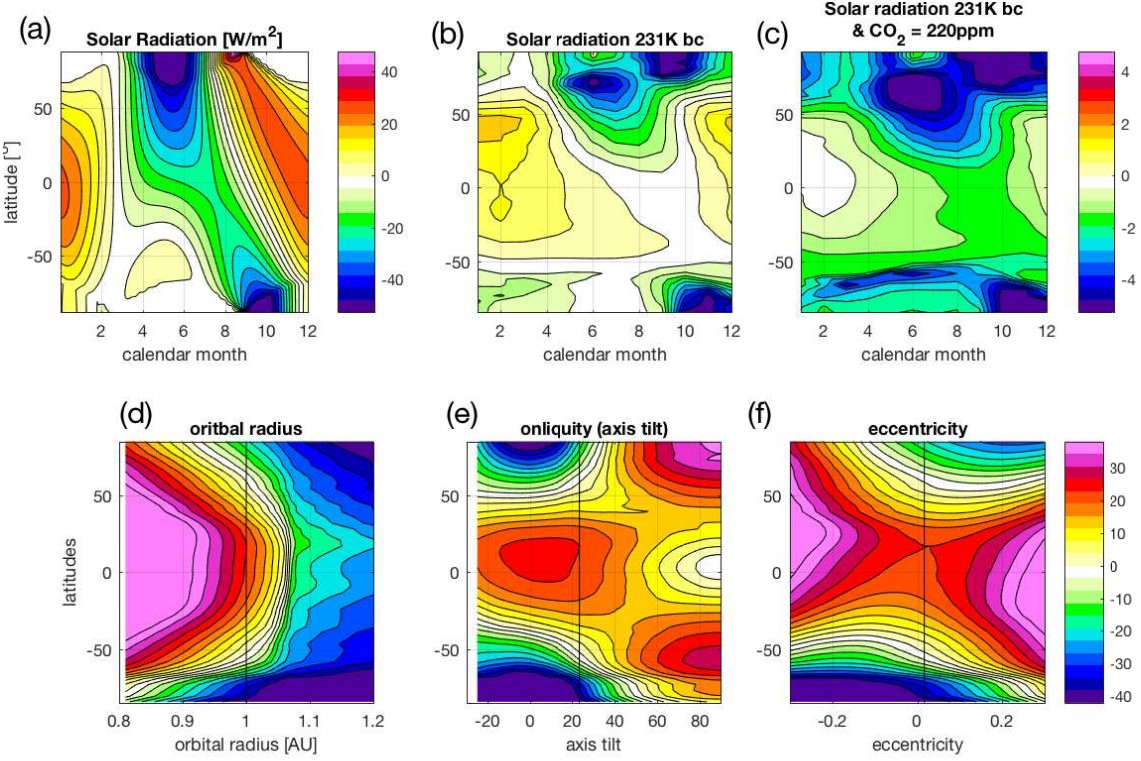

Figure 14: Orbital parameter forcings and Tsurf responses: (a) incoming solar radiation changes in the Solar-231Kyr experiment relative to the control GREB model. Tsurf response in Solar-231Kyr (b) and Solar-231Kyr-200ppm (c) relative to the control GREB model. Annual mean Tsurf in Orbit-radius (d), Obliquity (e) and Eccentricity (f). The solid vertical line in (d)-(f) marks the control (today) GREB model.