# Peer review of "The Monash Simple Climate Model"

_Geoscientific Model Development, 2018_

## Referee Comment (RC1) · Anonymous Referee #1 · 6 Sep 2018

Review of "The Monash Simple Climate Model Experiments (MSCM-DB v1.0): An interactive database of mean climate, climate change and scenario simulations", by Dommenget et al., submitted to Geoscientific Model Development.

Major Comments:

The authors propose the Monash Simple Climate Model experiment database for understanding climate processes for controlling mean climate, as well as how model climate in response to changes in CO2 or solar radiation forcings. It is an informative and

interesting experiment database and I can see the value of it. Therefore, I recommend the manuscript for publication after the authors address the following comments.

While it is understandable to use a simple model to understand the key processes that controls the climate and their response to different forcings, there are still limitations of what this simple model can achieve compared to the fully coupled global climate models or earth system models. I think it is important to discuss in details for the mean temperature or its seasonal cycle in response to certain processes that are significantly different from observations or previous GCM studies, at least for the processes discussed in this paper. For example, the cloud feedbacks are much more complicated in the full GCMs or in the real world. There is even large uncertainty from observations.

As the authors also pointed output, the model dynamics are not fully resolved in this energy balance model framework. The authors tried to comment on some of the drawback in the simulations because of lacking model dynamics, such as the midlatitude heat transport due to baroclinic waves. Similar issues of heat and momentum transport in the ocean are also present in this simple model configuration. Therefore, a more detailed discussion on how the mean climate or climate response would be without considering these dynamics in the atmosphere and ocean.

Another issue is using the word "observed" in many places in the text and figures. Unless I am mistaken, all these "observed" fields are still model simulations. It is misleading to use the word and I suggest to use something like "control" simulations to avoid confusion.

Detailed Comments:

1. Line 36, uncertainties of what?

2. Line 38, 10 degree C of surface temperature?

3. Lines 267-273, so, there is no other topography effect in this type of simple model simulations other than the effect on emissivity or $CO_2$ concentration?

[Figure]

4. Line 364, the eccentricity from 0.3 to 0.3?

5. Lines 429-432 and 496-499, I am not sure I understand why the strong cooling is due to the water vapour feedback. Is it because the water vapour is much less over the desert or mountain regions so that the warming effect due to water vapour is reduced.

6. Line 473, what is "it" that dampens the seasonal cycle.

7. Line 532, what do you mean by slow down the seasonal cycle?

8. Figure 11c, what are the red line and blue line? It's not explained in the caption.

---

## Referee Comment (RC2) · Anonymous Referee #2 · 5 Oct 2018

1) I think the major focus of this paper is more about to provide a simple GCM model output dataset for outreach purpose and less about model development and researches issue. I strongly suggest that this paper should be submitted to other journals or reports more focusing on dataset sharing or downstream applications. It also looks to me that present version of this paper is more like a report style for documenting purpose of the simple model experiments and datasets. It seems not a research article suitable for GMD. 2) Surface air temperature turns out to be the only climate variable in the model experiment dataset and the model tool and interactive webpage

seems more useful for other application fields such as policy making, heat-wave, and agriculture as well as social-economical impacts resulted from air temperature change under different warming scenarios (using different $CO_2$ concentration in the simulations of this dataset). Therefore, it looks to me that the dataset is more suitable published in other more relevant journals. 3) Abstract could be more specific in delivering the advantages and limitations of the experimental datasets. Moreover, the authors could elaborate more on their major findings from the thousand runs via using the simple model to draw the attention of readers for understanding how it can help with their studies. 4) (Section 2) It seems strange that GREB actually did flux corrections to constrain the model results close to observed mean climate while the focus of the model design and dataset is put on comparing mean climate. Moreover several parameters are input from climatological values e.g. cloud cover. Such strong constraints from climatological inputs will render the applications of the simple model for future prediction under global warming even the authors just care about air temperature. 5) The lack of considering circulation and cloud feedback in the GREB model is a big concern for climate model prediction. This limitation seems render the applications of the GREB for (2) the response of the climate to a doubling of the $CO_2$ concentration, and (3) scenarios of external $CO_2$ concentration and solar radiation forcings as discussed in the manuscript. 6) (Mean climate) Clouds and hydrological cycle turn out to be the two most important factors as shown in controlling the annual mean as shown in Figure 7. However, these two major factors are highly related to cloud and precipitation processes which are not explicitly simulated in the atmospheric layer of present model. Also, I am wondering how the GREB model deals with precipitation. I guess it is also from reanalysis model output. I think these missing processes will significantly affect the estimation of air temperature under global warming via setting different $CO_2$ concentrations. 7) More relevant references from comprehensive GCMs to backup the findings of figure 7 or discussions regarding to mean climate can increase the scientific merit of the present version as the authors did for double $CO_2$ and scenarios simulation part. Also, the comparisons to previous literatures mentioned in the double $CO_2$

and scenarios part could be more detailed e.g. more discussions on sources of uncertainties from the usage of the simple model versus the comprehensive GCMs. 8) I agree that such simple model for air temperature simulation can be useful for rough estimation purpose or primary understanding of the role of possible processes but not so applicable for the future climate projections. Similar to my concern 1), I also suggest that probably more high horizontal resolution version of the GREB experimental simulations can be more useful for other communities interest about effects associated with increase of temperature.

---

## Author Comment (AC1) · 15 Nov 2018

**Revisions of "The Monash Simple Climate Model Experiments (MSCM-DB v1.0): An interactive database of mean climate, climate change and scenario simulations"**

Dear Editor and referees,
we like to thank the referees and editor for the time spend on reviewing this manuscript and for the many very helpful comments they provided. We think the referee comments have helped us to substantially improve the presentation of this work. Below we give a point-to-point response to all referee comments, hoping the revised manuscript has now been improved in clarity and is ready for publication.

With best regards,

Dietmar Dommenget, Kerry Nice, Tobias Bayr, Dieter Kasang, Christian Stassen and Mike Rezny

*Major Comments:*
*The authors propose the Monash Simple Climate Model experiment database for understanding climate processes for controlling mean climate, as well as how model climate in response to changes in CO2 or solar radiation forcings. It is an informative and interesting experiment database and I can see the value of it. Therefore, I recommend the manuscript for publication after the authors address the following comments.*

**Response:** We like to thank the referee for the evaluation of our manuscript and the comments that will help us to improve the model. See detailed responses below.
* * *
*While it is understandable to use a simple model to understand the key processes that controls the climate and their response to different forcings, there are still limitations of what this simple model can achieve compared to the fully coupled global climate models or earth system models. I think it is important to discuss in details for the mean temperature or its seasonal cycle in response to certain processes that are significantly different from observations or previous GCM studies, at least for the processes discussed in this paper. For example, the cloud feedbacks are much more complicated in the full GCMs or in the real world. There is even large uncertainty from observations.*

**Response:** We revised the manuscript to better discuss some of these aspects. We do point out some of the limitations several times in the manuscript. However, we need to keep in mind the space limitations within this journal and can therefore not go into all details.
The cloud feedbacks are indeed important, much more complex and uncertain. We therefore think it is really beyond this paper to discuss this appropriately and have to leave it by saying that the GREB model cannot simulate these.
* * *
*As the authors also pointed output, the model dynamics are not fully resolved in this energy balance model framework. The authors tried to comment on some of the drawback in the simulations because of lacking model dynamics, such as the midlatitude heat transport due to baroclinic waves. Similar issues of heat and momentum transport in the ocean are also present in this simple model configuration. Therefore, a more detailed discussion on how the mean climate or climate response would be with- out considering these dynamics in the atmosphere and ocean.*

**Response:** We think this is related to the above comment. We revised the manuscript to better discuss some of these aspects, but again we need to point out that it is beyond this paper to give a full discussion of all these aspects.
* * *
*Another issue is using the word "observed" in many places in the text and figures. Unless I am mistaken, all these "observed" fields are still model simulations. It is misleading to use the word and I suggest to use something like "control" simulations to avoid confusion.*

**Response:** We do compare here to the observed. The surface temperature in observations and

the control simulation are identical by construction, due to the flux correction terms and lag of internal variability. This is different from CGCM simulations. Therefore, when we show the observed Tsurf, it is the same as the control simulation of the GREB model. We made some changes to the figure caption of Fig. 4 to improve the clarity.
* * *
*Detailed Comments:*

*1. Line 36, uncertainties of what?*

**Response:** We revised the sentence.
* * *
*2. Line 38, 10 degree C of surface temperature?*

**Response:** Yes! We included surface temperature in the text.
* * *
*3. Lines 267-273, so, there is no other topography effect in this type of simple model simulations other than the effect on emissivity or CO2 concentration?*

**Response:** We indeed forgot to mention that the topography also affects the diffusion coefficient for the transport of heat and moisture. This is now stated in the text. It has no discernible effect on the results that we discussed in this study and therefore we forgot to mention it.
The wind field is otherwise not affected by topography as we are prescribing the wind field and changes in the wind field regarding the topography would require a GCM approach, which the GREB model does not simulate.
* * *
*4. Line 364, the eccentricity from 0.3 to 0.3?*

**Response:** Yes! It does sound strange, but eccentricity is between 0 to 1; it has no negative values. But with earth axis tilt (earth rotating around itself) relative to the earth-sun orbit plane or relative to our monthly calendar, it does matter what orientation the orbit has. Therefore, we stated "(Earth closest to the sun in July)".
* * *
*5. Lines 429-432 and 496-499, I am not sure I understand why the strong cooling is due to the water vapour feedback. Is it because the water vapour is much less over the desert or mountain regions so that the warming effect due to water vapour is reduced.*

**Response:** Hmm, yes and no. The response of the climate system to any external forcing or change in boundary conditions is dominated by internal positive feedbacks. The most important positive feedback is the water vapor feedback, and, yes, the much less water vapor in deserts and mountain regions will make those regions more sensitive to the water vapor feedback. Thus, the water vapor feedback is stronger here.
Our text was indeed not clear enough to explain this properly. We tried to extend the text in

this passage to better highlight this.
* * *
*6. Line 473, what is "it" that dampens the seasonal cycle.*

**Response:** The hydrological cycle. We revised the text.
* * *
*7. Line 532, what do you mean by slow down the seasonal cycle?*

**Response:** Slow down is indeed a bit confusing. We now say "reduce".
* * *
*8. Figure 11c, what are the red line and blue line? It's not explained in the caption.*

**Response:** They are two different experiments, which are now mentioned in the figure caption and also listed in Table 3.
* * *
*Referee #2*

*1) I think the major focus of this paper is more about to provide a simple GCM model output dataset for outreach purpose and less about model development and researches issue. I strongly suggest that this paper should be submitted to other journals or reports more focusing on dataset sharing or downstream applications. It also looks to me that present version of this paper is more like a report style for documenting purpose of the simple model experiments and datasets. It seems not a research article suitable for GMD.*

**Response:** The MSCM database has some teaching aspects and may potentially also be useful for outreach. However, the focus of this work is on the research aspects of this database. We therefore think the GMD journal is the best journal for this work. From our perspective, a paper that focus on "outreach" would be very different from the study that we presented.
We tried to revise the presentation the best we could to better high-light the research value to this database. Please, see also our response to the other comments.
* * *
*2) Surface air temperature turns out to be the only climate variable in the model experiment dataset …*

**Response:** The GREB does simulate more than just the surface temperature. It simulates four prognostic variables: surface, atmospheric and subsurface ocean temperature, and atmospheric humidity (column integrated water vapor). It further simulates a number of diagnostic variables, such as precipitation and snow/ice cover.
We now explicitly state this in the model section 2 and in the code availability section 5.
* * *
*… and the model tool and interactive webpage seems more useful for other application fields such as policy making, heat-wave, and agriculture as well as social-economical impacts resulted from air temperature change under different warming scenarios (using different CO2 concentration in the simulations of this dataset). Therefore, it looks to me that the dataset is more suitable published in other more relevant journals.*

**Response:** We think that the model experiments described here are primarily of interest to climate scientists. The three sets of experiments that we discuss (mean state, climate change and scenarios) are primarily focused on understanding the physical processes of the climate system. The focus is on how different climate processes interact to create the climate as we know it and how it would respond to external forcing.
A climate model for policy making, agriculture or social-economical impact studies would probably not focus so much on the physical climate process interactions, but more on the impact of climate. But these are not simulated in these GREB model experiments. An example for such a model would be the MAGICC climate model, which aims at fast simulations of different climate change scenarios. It does not simulate the details of the physical processes as the GREB model does.
While the GREB model maybe useful for such studies, it is not the aim of this study. We hope that the revised manuscript does make it clear that this is a study or database for the physical understanding of the climate system.

*3) Abstract could be more specific in delivering the advantages and limitations of the experimental datasets. Moreover, the authors could elaborate more on their major findings from the thousand runs via using the simple model to draw the attention of readers for understanding how it can help with their studies.*

**Response:** We changed the abstract to better guide the reader in what these model experiments are useful for. However, we have to keep in mind that the space limitations in this journal and can therefore not elaborate much about the findings of all of these experiments. The main aim of this study is to give an overview about the scientific robustness and limitations of the database, but not to discuss the results in each of these experiments.

*4) (Section 2) It seems strange that GREB actually did flux corrections to constrain the model results close to observed mean climate while the focus of the model design and dataset is put on comparing mean climate. Moreover, several parameters are input from climatological values e.g. cloud cover. Such strong constraints from climatological inputs will render the applications of the simple model for future prediction under global warming even the authors just care about air temperature.*

**Response:** The model indeed uses flux correction in some of the experiments, but not in the ones we use to discuss the mean state climate. The referee may have overlooked this. The experiments discussed in section 3a,b do not use flux corrections. We have explicitly stated this in section 3a and now also state it again in section 3b. It is also mentioned in the figure captions.
In some experiments flux correction are useful when changes are considered small, such as the response to increased CO2 concentrations. Therefore, the response to 2xCO2 forcing and some of the scenarios use flux corrections. This assures that the response discussed are relative to the observed control climate. This is the same approach as in DF11.
The limitation of the GREB model in not simulating the atmospheric circulation nor the cloud cover formation is important, and indeed limits the results of the GREB model experiments. We have made these limitations clear in the manuscript. We hope that the revised manuscript does give a fair representation of the GREB model's skill and limitations.

*5) The lack of considering circulation and cloud feedback in the GREB model is a big concern for climate model prediction. This limitation seems render the applications of the GREB for (2) the response of the climate to a doubling of the CO2 concentration, and (3) scenarios of external CO2 concentration and solar radiation forcings as discussed in the manuscript.*

**Response:** We agree with the referee. This is why we think the main aim of this database is a conceptual understanding and a first guess. It should not be considered as a best guess for future climate change projections. It does not replace or improve the projections of CGCM

simulations as such.

We revised the manuscript to better discuss some of these limitations and illustrate the purpose of this database. See also our reply to a similar comment about the role of the atmospheric circulation and cloud feedback from referee one.
* * *
*6) (Mean climate) Clouds and hydrological cycle turn out to be the two most important factors as shown in controlling the annual mean as shown in Figure 7. However, these two major factors are highly related to cloud and precipitation processes which are not explicitly simulated in the atmospheric layer of present model. Also, I am wondering how the GREB model deals with precipitation. I guess it is also from reanalysis model output. I think these missing processes will significantly affect the estimation of air temperature under global warming via setting different CO2 concentrations.*

**Response:** The GREB model does simulate the hydrological cycle including precipitation. This is stated in section 2, but may have been missed by the referee. The hydrological cycle is indeed one of the most important aspects of the climate system and is therefore an important process that a climate model needs to simulate. This is why the GREB model does simulate this process. The atmospheric humidity is a prognostic variable (eq.A4) and precipitation is simulated in respect to the atmospheric humidity, see DF11.

The cloud cover is also simulated in terms of its impact on short and long wave radiation. These are the mean effects it has in the context of the mean climate. Cloud feedbacks, that is, changes in response to the climate, are indeed not simulated and are a limitation of the model. We tried to improve the presentation of manuscript to better reflect these limitations.
* * *
*7) More relevant references from comprehensive GCMs to backup the findings of figure 7 or discussions regarding to mean climate can increase the scientific merit of the present version as the authors did for double CO2 and scenarios simulation part. Also, the comparisons to previous literatures mentioned in the double CO2 and scenarios part could be more detailed e.g. more discussions on sources of uncertainties from the usage of the simple model versus the comprehensive GCMs.*

**Response:** We do acknowledge the referees need for more reference from *comprehensive GCMs to backup the findings.* We therefore did add a bit more discussion of these results in respect to some previous publications in section 3b. However, we have to keep in mind the limitations within this format and the aim of the study to only introduce this database. More in-depth discuss must be left for future studies.
* * *
*8) I agree that such simple model for air temperature simulation can be useful for rough estimation purpose or primary understanding of the role of possible processes but not so applicable for the future climate projections. Similar to my concern 1), I also suggest that probably more high horizontal resolution version of the GREB experimental simulations can be more useful for other communities interest about effects associated with increase of temperature.*

**Response:** The focus of this study is indeed on the physical process in the climate system and

the understanding of their interactions on the large scale. We think that detailed future climate change projections, in particular on higher regional resolutions are not the main application of this database. This model is more for fast first guesses and conceptual understanding. We hope that the revised manuscript does make this point. In particular, we tried to improve the abstract and summary section to highlight this.

---

## Author Response (AR2)

**Revisions of "The Monash Simple Climate Model Experiments (MSCM-DB v1.0): An interactive database of mean climate, climate change and scenario simulations"**

Dear Editor and referees,
we like to thank the referees and editor for the time spend on reviewing this manuscript again. We are sorry that some comments appear to be insufficiently responded to. Below we give a point-to-point response to all remaining referee comments, hoping the revised manuscript will now be ready for publication.

With best regards,

Dietmar Dommenget, Kerry Nice, Tobias Bayr, Dieter Kasang, Christian Stassen and Mike Rezny

*Major Comments:*
*The authors address some of my previous comments. However, there are still missing information for the manuscript to be addressed.*

*It is NOT an excuse to use the limitation of the length of the journal for not to explain the information clearly when the reviewer clearly asks the authors to. The authors can always make other part of the manuscript concise. Also, the authors cannot just say that this is beyond the scope of this manuscript. If both reviewers raised the same concern, the authors should address the issues.*

**Response:** Please see our respond below. We hope this now does give an appropriate response.
* * *
*The major issue is still regarding the lack of discussion of feedbacks and interactions of clouds, circulation and also aerosols within the climate system in this particular model configuration since the "2xCO2 response deconstruction" and "Scenarios" are the two main foci of the manuscript. Since many research literatures are focusing on the feedbacks of these processes in response to changes in CO2 in the fully coupled GCMs, the authors should include the discussion of what the readers or users of this model should or should not expect the changes in temperature in response to different forcings when these processes (clouds, circulation and aerosols) are not included in the simulations.*

*The authors MUST address this particular issue or I cannot recommend for publication.*

**Response:** We are sorry that it appears that we have not addressed this important issue in our first revisions. Our first respond (in the previous revisions) to this comment was indeed a bit too short and it did not fully reflect that we have indeed included a number of changes in the manuscript. It is also related to the comment 5 of referee #2 (in the previous revisions).
We made a number of changes in the manuscript to better discuss the circulation and cloud feedbacks. In the combined first and current revision we made the following changes:

(*) Introduction: We now explicitly state that the GREB model does not simulate cloud and circulation feedbacks.

(*) Model and experiment descriptions: We now state in the first part of this section that experiments of this database neglect any effects resulting from cloud or circulation feedbacks and that some of these limitations will be discussed in the context of the results.

(*) Section 3b: We extended the discussion of the cloud effect and how feedbacks in the climate system will affects this. We further added a new full paragraph to point out how cloud and circulation feedbacks are likely to alter some of the results in sections 3b.

(*) Summary and discussion: At the end of this section we again point out the limitations of the GREB model in respect to cloud and circulation feedbacks.

The referee now also mentioned the effect of aerosols, which was not mentioned in the first review. Aerosols are not simulated in the GREB model and no experiment is related to the effects of aerosols. However, we do mention at the end of the summary that including aerosols should be a focus for future development.
* * *
*Detailed Comments*

*1. Line 35, this sentence is still not clear. Which mean climate processes have uncertainties on the order of 20-30%? A table for a list of processes showing the number of uncertainties of specific climate processes is necessary.*

**Response:** We decided to delete this sentence from the abstract as it is indeed a bit unclear. It is related to the result discussed in section 3a: "*The GREB model performance can be put in perspective by illustrating how much the climate processes simulated in the GREB model contribute to the mean climate relative to the bare world simulation (see* Fig. 4*). …*"
We think the discussion in section 3a is ok as it is and it explains where the 20-30% comes from. A detailed study or table of how each process is uncertain may in principal be possible, but we think that this would take much more space and it is not what we aim for here. It also needs to be noted that non-linearities and missing processes would need to be discussed here too.
* * *
*2. Line 589, the sentence is still confusing. What aspect(s) of the seasonal cycle (amplitude?) is reduced by the ocean?*

**Response:** We revised the sentence. The large ocean heat capacity reduces the amplitude of the seasonal cycle.
* * *
**Referee #2**

*accepted as is.*

**Response:** We like to thank the referee for accepting the current version.
* * *

[revised manuscript text omitted]

---

## Author Response (AR3)

**Revisions of "The Monash Simple Climate Model Experiments (MSCM-DB v1.0): An interactive database of mean climate, climate change and scenario simulations"**

Dear Editor and referee,
Thank you for evaluating this manuscript again. We are really sorry that the referee is still not happy with the manuscript. In the previous revisions we made a number of changes to the manuscript to address the referee's comments, but now we do not really see how we can further improve the manuscript in response to the referee's comment on cloud feedback or model limitations. Below we give a point-to-point response to the referee comments, that explain why we think we have already addressed the referee's comments. We hope that the referee will see that we have indeed taken his/her comments on board and that we have included them in the manuscript.

With best regards,

Dietmar Dommenget, Kerry Nice, Tobias Bayr, Dieter Kasang, Christian Stassen and Mike Rezny

**Referee #1**

*Major Comments:*
*The authors still did not address my comments properly. I compared the latest revision (V7) and the last version (V6). The only paragraph added in 3b is the following:*

*In the above discussion on how the individual climate processes affect the climate we have to keep in mind the limitations of the GREB model and the experimental setups. The changing a single climate element is more complex in the real world than simulated in these GREB experiments. For instance, if the ocean heat capacity is turned 'OFF' it will n effective heat capacity, but the resulting changes in surface temperature gradients will also affect the atmospheric circulation patterns and subsequently the cloud cover. Such circulation and cloud cover are neglected in the GREB model, as they are given as fixed boundary conditions. Regionally such effects can be significant and CGCM simulations a effects.*

**Response:** The referee is indicating that the only changes we introduced in the response to his/her comments on the limitations of the model and the cloud feedback is the paragraph mentioned above. However, we have made a number of changes to the original manuscript in response to the referee's comments. Below we will summarise all the changes that we have introduced from the first submission to the last revision:

"**1. Introduction**
…
… It further also does not include cloud feedbacks or adjustments in the atmospheric circulation, as both are given as boundary conditions. However, it does include the most important water vapor, black-body radiation and ice-albedo feedbacks. …"

"2. Model and experiment descriptions
…
… . Subsequently, the experiments of this database neglect any effects resulting from cloud or circulation feedbacks. These experiments should therefore only be considered as first guess estimates. In some aspects of these experiments the missing feedbacks and processes will be important. In the context of some of the results we will discuss some of these limitations.
…
**b. Mean climate deconstruction**
…
In the discussion of the experiments, it is important to consider that climate feedbacks are

contributing to the interactions of the climate processes. The effect of a climate process on the climate is a result of all the other active climate processes responding to the changes that the climate process under consideration introduces. It also depends on the mean background climate. Therefore, it does matter in which combination of switches the GREB model experiments are discussed. For instance, the effect of the Ice/Snow cover, is stronger in a much colder background climate, but is also affected by the feedback in other climate processes, such as the water vapour feedback. We will therefore consider different experiments or different experiment sets to shade some light into these interactions.

…

The cloud cover in the GREB model is only considered as a given boundary condition, but does not simulate the formation of clouds. Therefore, it does not include cloud feedbacks. However, the mean cloud cover does influence the radiation balance and therefore affects the mean climate and its seasonal cycle. …

…

In the above discussion on how the individual climate processes affect the climate we have to keep in mind the limitations of the GREB model and the experimental setups. The climate response to changing a single climate element is more complex in the real world than simulated in these GREB experiments. For instance, if the ocean heat capacity is turned 'OFF' it will not just have an effect on the effective heat capacity, but the resulting changes in surface temperature gradients will also affect the atmospheric circulation patterns and subsequently the cloud cover. Such effects on the atmospheric circulation and cloud cover are neglected in the GREB model, as they are given as fixed boundary conditions. Regionally such effects can be significant and CGCM simulations are required to study such effects."

"**4. Summary and discussion**

…

… . Here we need to keep in mind the limitation that the GREB model does not consider atmospheric or ocean circulation changes nor does it simulate cloud cover feedbacks. Such processes will alter this picture somewhat and need to be studied with more complex climate models, which may in particular be important for more detailed regional information of future climate change or social-economical impact studies."

In summary, we think that we made significant revisions of our manuscript to address the referee's comments. The referee seems to be indicating that he/she likes to see more discussions of literature related to circulation changes or cloud feedbacks(?) for section 3b. Section 3b is about the "Mean climate deconstruction". The references the referee is suggesting are really not directly related to these experiments. Circulation changes and cloud feedbacks are discussed in the context of CO2 forcing in the literature, but not for the experiments that we discuss here. We therefore think that our discussion at the end of this subsection does provide a good way of incorporating these problems and point the reader to the limitations of these simulations.

*My original question remains the same:*

*The authors should include the discussion of what the readers or users of this model should or should not expect the changes in temperature in response to different forcings w circulation and aerosols) are not included in the simulations.*

**Response:** We think that the current manuscript gives a very good discussion on what the "*readers or users of this model should or should not expect the changes in temperature in response to different forcings*". We really can't see how this could be improved or why the referee thinks that we have not provided an appropriate discussion.

Throughout the manuscript we carefully point out the limitations of the GREB model. We would like to quickly summarise what we have already included in the manuscript:

"1. Introduction

… The GREB model differs, in that it follows an energy balance approach and does not simulate the geophysical fluid dynamics of the atmosphere. It is therefore a climate model that does not include weather dynamics, … . It further also does not include cloud feedbacks or adjustments in the atmospheric circulation, as both are given as boundary conditions.  …"

"2. Model and experiment descriptions

… Thus, the GREB model does not simulate the atmospheric or ocean circulation and is therefore conceptually very different from CGCM simulations.

… , but an important limitation of the GREB model is that the response to external forcings or model parameter perturbations do not involve circulation or cloud feedbacks, which are relevant in CGCM simulations [Bony et al. 2006]. Subsequently, the experiments of this database neglect any effects resulting from cloud or circulation feedbacks. These experiments should therefore only be considered as first guess estimates. In some aspects of these experiments the missing feedbacks and processes will be important. In the context of some of the results we will discuss some of these limitations."

"a. **Experiments for the mean climate deconstruction**

**Hydrological cycle**:

…
It needs to be noted here, that the atmospheric emissivity in the log-function parameterization of eq. [A9] can become negative, if the hydrological cycle, cloud cover and $CO_2$ concentration are switched OFF (set to zero). This marks an unphysical range of the GREB emissivity function and we will discuss the limitations of the GREB model in these experiments in Section 3b.

…

**Model Corrections:**

…

It should be noted here that the model correction terms in the GREB model have been introduced … . They are meaningful for a small perturbation in the climate system, but are less likely to be meaningful when large perturbations to the climate system are done (e.g. cloud cover set to zero)."

**"3. Some results of the model simulations**
…

**a. GREB model performance**
[the whole section]

…

**b. Mean climate deconstruction**
…
… However, we need to consider that the experiment with switching OFF the hydrological cycle is the only experiment in which we have a significant amount of global cooling (by about -44°C). As a result, most of the earth is below freezing temperatures and therefore has a much stronger ice-albedo feedback than in any other experiment. This leads to a significant amplification of the response.
It is instructive to repeat the experiments with the ice-albedo feedback switched OFF …
…
… However, as mentioned in the appendix A1 the log-function approximation leads to negative emissivity if all greenhouse gasses ($CO_2$ and water vapour) concentrations and cloud cover are zero. The negative emissivity turns the atmospheric layer into a cooling effect, which dominates the impact of the atmosphere in this experiment (Figs. 8b, c). This is a limitation of the GREB model and the result of this experiment as such should be considered with caution.
…
…
In the above discussion on how the individual climate processes affect the climate we have to keep in mind the limitations of the GREB model and the experimental setups. The climate response to changing a single climate element is more complex in the real world than simulated in these GREB experiments. For instance, if the ocean heat capacity is turned 'OFF' it will not just have an effect on the effective heat capacity, but the resulting changes in surface temperature gradients will also affect the atmospheric circulation patterns and subsequently the cloud cover. Such effects on the atmospheric circulation and cloud cover are neglected in the GREB model, as they are given as fixed boundary conditions. Regionally such effects can be significant and CGCM simulations are required to study such effects."

**"4. Summary and discussion**
… The GREB model is a simple climate model that does not simulate internal weather variability, circulation, or cloud cover changes (feedbacks). …
…
The GREB model without flux corrections simulates the mean observed climate well and has an uncertainty of about 10°C. The model has larger cold biases in the polar regions indicating that the meridional heat transport is not strong enough. Relative to a bare world without any climate processes the RMSE is reduced to about 20-30% relative to observed. Further, the

GREB models emissivity function reaches unphysical negative values when water vapour, $CO_2$ and cloud cover is set to zero. This is a limitation of the log-function parametrization, that can potentially be revised if a new parameterization is developed that considers these cases. …

…

… Here we need to keep in mind the limitation that the GREB model does not consider atmospheric or ocean circulation changes nor does it simulate cloud cover feedbacks. Such processes will alter this picture somewhat and need to be studied with more complex climate models, which may in particular be important for more detailed regional information of future climate change or social-economical impact studies.

…"

We really think that this is a quite extensive discussion of the model limitations and the reader will indeed be well informed on the limitations of these simulations. If the referee still thinks that this is not the case, we would appreciate a more specific comment on what aspect, in which subsection, we may have been misleading the reader by not pointing out the limitations of these simulations correctly. We simply can't see what has not been covered sufficiently.
* * *
*There are so many literatures discussing cloud feedbacks. The last IPCC report even have a chapter for clouds (https://archive.ipcc.ch/pdf/assessmentreport/ar5/wg1/WG1AR One of the WCRP grand challenges also focuses on clouds, circulation and climate sensitivity (https://www.wcrpclimate.org/gcclouds). The authors can start from here and re to have a better discussion to address my comments.*

**Response:** The referee asks us to discuss more on "*cloud feedbacks*". We do not understand why, in which context, or what part of the manuscript has not addressed this. We simply can't see how we would improve the presentation by discussing more on "*cloud feedbacks*" other than what we have already included in the manuscript. In what subsection would more discussion on cloud feedbacks be useful? We simply do not see that.

The references/discussion the referee is pointing out is about cloud feedbacks in the context of climate sensitivity (e.g. IPCC report and related references), thus in the context of CO2-forcing. The vast majority of the experiments that we discuss are not directly related to these experiments (e.g. section 3a, b and d). Only section 3c ("2xCO2 response deconstruction") and partly 3d are somewhat related to this. However, here we only discuss aspects of the response pattern with specific limitations (e.g. isolated ice-albedo or water vapor feedbacks). It does not address cloud feedbacks or the connection to this fairly loose.

The reference DF11, on which this current paper is based on, discusses the GREB response to CO2-forcing. In the context of DF11 a more detailed discussion of cloud feedbacks may seem appropriate, but we can't rewrite the DF11 reference in this manuscript. This would seem inappropriate to us. We therefore think we have to limit this discussion to the caveat notes that we have already made very clearly in the text.
* * *
*Again, I am returning it back for another round of major revision. The authors MUST address this particular issue or I CANNOT recommend for publication. Also, the authors sh change*

*version of the revised manuscript showing what paragraphs are added to address my comments.*

**Response:** We are sorry, but without more specific comments about what we have not discussed sufficiently in the text (see our responses above), we are unable to make major revision to the manuscript. The only change that we have made to the text is to include a few more references on cloud feedbacks in section 2; 3$^{rd}$ paragraph. We will be happy to make further revisions if we are provided more detailed guidance on what hasn't already been addressed.

---

## Author Response (AR4)

**Revisions of *"The Monash Simple Climate Model Experiments (MSCM-DB v1.0): An interactive database of mean climate, climate change and scenario simulations"**

Dear Editor,

Thank you for evaluating this manuscript again. We addressed all of your comments. Below we give a point-to-point response to your comments. We hope the manuscript is now ready for publication.

With best regards,

Dietmar Dommenget, Kerry Nice, Tobias Bayr, Dieter Kasang, Christian Stassen and Mike Rezny

*Editor*

*Topical Editor Decision: Publish subject to minor revisions (review by editor) (02 Apr 2019) by Min-Hui Lo*
*Comments to the Author:*

*Dear Authors,*
*I have couples comments for your revised manuscript:*

**Response:** Please see our response to all comments below.

*1.     Can you acknowledge the issue of missing cloud feedback and its consequence on the abstract? I would also suggest that you can include couple sentences based on those references that you cited (about the cloud feedbacks) to illustrate the importance of cloud feedbacks or what the impacts could be without considering the cloud feedbacks.*

**Response:** We now state the missing cloud feedback in the abstract and also added a few more words to the cloud feedbacks in the modelling section where we cited the related references.

*2.     Line 151: "In some aspects of these experiments",*
*please be more specific.*

**Response:** We deleted the sentence, as it is indeed a bit too vague.

*3.     Line 416: "It seems likely that the meridional heat transport is the main limitation in the GREB model, given the too warm tropical regions and the, in general, too cold polar regions and the too strong seasonal cycle in the polar regions in the GREB model without correction terms."*

*You nicely mentioned the impacts of the meridional heat transport. Can you also indicate the consequence of lacking the cloud feedback on the simulations?*

**Response:** We don't think a discussion of cloud feedback would make sense in this context. In these experiments we discuss the simulation of the mean state by the processes simulated in GREB. The cloud cover is given as a boundary condition, but formation of clouds is not a process simulated in this model. Cloud feedbacks can therefore not contribute to limitations in the simulation of the mean state, as the meridional heat transport can.
* * *
*4. Line 463-466: "Previous studies on the cloud cover effect on the overall climate mostly focus on the radiative forcings estimates, but to our best knowledge do not present the overall change in surface temperature [e.g. Rossow and Zhang 1995]."*

*While it does not significantly affect the surface temperature, the lack of cloud feedbacks can affect the radiative forcings estimates. Thus, can you elaborate it more on how the radiative forcings might be affected?*

**Response:** There may be some misunderstanding here. Clearly the mean cloud cover does affect the mean surface temperature. However, previous studies have not quantified by how much, as they did not conducted simulations as discussed in this study. They therefore only discussed by how much the radiation is affected. We slightly changed the wording in this paragraph to better highlight this.
* * *
*5. Line 825:826: "need to be studied with more complex climate models,"*

*is it possible to study this using the GREB model? And how?.*

**Response:** We added a few lines to discuss this. The current model does not allow for circulation or cloud cover changes in response to external forcings. However, the structure of the GREB model would allow to include such models.
* * *

[revised manuscript text omitted]